# Operando probing of the surface chemistry during the Haber–Bosch process

Christopher M. Goodwin[1,5 ✉], Patrick Lömker[1], David Degerman[1], Bernadette Davies[2], Mikhail Shipilin[1], Fernando Garcia-Martinez[3], Sergey Koroidov[1], Jette Katja Mathiesen[1], Raffael Rameshan[4], Gabriel L. S. Rodrigues[1], Christoph Schlueter[3], Peter Amann[1,6] & Anders Nilsson[1 ✉]

The large-scale conversion of $N_2$ and $H_2$ into $NH_3$ (refs. 1,2) over Fe and Ru catalysts[3] for fertilizer production occurs through the Haber–Bosch process, which has been considered the most important scientific invention of the twentieth century[4]. The active component of the catalyst enabling the conversion was variously considered to be the oxide[5], nitride[2], metallic phase or surface nitride[6], and the rate-limiting step has been associated with $N_2$ dissociation[7–9], reaction of the adsorbed nitrogen[10] and also $NH_3$ desorption[11]. This range of views reflects that the Haber–Bosch process operates at high temperatures and pressures, whereas surface-sensitive techniques that might differentiate between different mechanistic proposals require vacuum conditions. Mechanistic studies have accordingly long been limited to theoretical calculations[12]. Here we use X-ray photoelectron spectroscopy—capable of revealing the chemical state of catalytic surfaces and recently adapted to operando investigations[13] of methanol[14] and Fischer–Tropsch synthesis[15]—to determine the surface composition of Fe and Ru catalysts during $NH_3$ production at pressures up to 1 bar and temperatures as high as 723 K. We find that, although flat and stepped Fe surfaces and Ru single-crystal surfaces all remain metallic, the latter are almost adsorbate free, whereas Fe catalysts retain a small amount of adsorbed N and develop at lower temperatures high amine ($NH_x$) coverages on the stepped surfaces. These observations indicate that the rate-limiting step on Ru is always $N_2$ dissociation. On Fe catalysts, by contrast and as predicted by theory[16], hydrogenation of adsorbed N atoms is less efficient to the extent that the rate-limiting step switches following temperature lowering from $N_2$ dissociation to the hydrogenation of surface species.

Figure 1a shows how surface-sensitive operando X-ray photoelectron spectroscopy (XPS) is measured together with reaction-product detection during the Haber–Bosch process in the POLARIS instrument[13]. XPS is a powerful technique for investigating the chemical state of catalytic surfaces through core-level shifts that traditionally required vacuum conditions, but operando studies can be conducted using a differential pumping scheme[17]. The Fe and Ru single-crystal surfaces are mounted in front of the electron spectrometer with a gap of 30 μm and gases are fed through the front cone of the electron lens, creating a localized virtual catalytic reactor of elevated pressure with a rapid gas flow[13]. The typical operational pressure for ammonia synthesis is 50–200 bar (ref. 18), at which the gas-phase equilibrium is strongly shifted towards the product, giving a high final conversion to ammonia. However, during the initial phase of the Haber–Bosch process, when not much ammonia has yet been produced, the reaction also proceeds with a high rate at our operational pressures of up to 1 bar (refs. 19,20).

The incoming X-rays were set to an energy of 4,600 eV and the incidence at an angle below total reflection, allowing for high surface sensitivity despite high kinetic energy electron detection. The emitted photoelectrons will pass into the spectrometer through orifices in the front cone and be detected in a hemispherical analyser. The inset in Fig. 1a shows an example of an N1s spectrum of 1:3 $N_2$:$H_2$ gases at 1 bar at 673 K, indicating $NH_3$ (blue), $NH_2$ (purple), NH (red), surface N (green) and nitride surface (yellow) components. The measurements were conducted at a photon flux at which no detectable X-ray-beam-induced changes could be seen during the Haber–Bosch process (see Methods for further details).

To track the production of $NH_3$, masses 15 and 16 were monitored in the mass spectrometer (see Methods), as shown in Fig. 1b. The relative chemical reactivities shown in Fig. 1c were determined by measuring the mass spectrometer ammonia signal with respect to the signal of all constituents to compute the number of ammonia molecules formed per second per surface site, which is then further normalized to the

[1]Department of Physics, Stockholm University, AlbaNova University Center, Stockholm, Sweden. [2]Department of Materials and Environmental Chemistry, Stockholm University, Stockholm, Sweden. [3]Photon Science, Deutsches Elektronen-Synchrotron DESY, Hamburg, Germany. [4]Institute of Physical Chemistry, Montan University Leoben, Leoben, Austria. [5]Present address: Materials Science, ALBA Synchrotron Light Facility, Cerdanyola del Vallés, Spain. [6]Present address: Scienta Omicron AB, Uppsala, Sweden. ✉e-mail: cgoodwin@cells.es; andersn@fysik.su.se

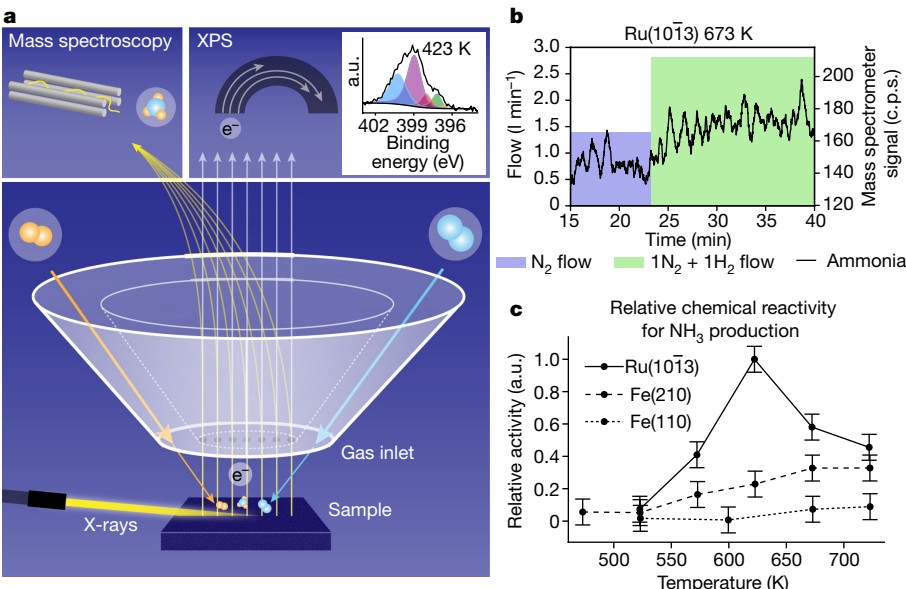

**Fig. 1 | Experimental set-up and relative turnover-frequency measurements. a**, The sample faces a set of apertures that deliver the reaction gas while simultaneously gathering products and emitted electrons. The grazing incidence X-rays enter from the left, producing photoelectrons. The mix of gas and electrons is separated by an electrostatic lens and analysed in an electron analyser and a mass spectrometer. The inset shows XPS spectra of the chemical state of N at 200 mbar over the Fe(110) surface with a 1:3 $N_2$:$H_2$ gas ratio. **b**, Mass

spectrometer readout of masses 15 and 16 corresponding to $NH_3$ production as the gas ratio changes from 150 mbar pure $N_2$ (blue region showing flow) to 300 mbar 1:1 $N_2$:$H_2$ (green region showing flow) over Ru at 673 K. Note that the flows of the gases are shown as the filled blocks plotted on the left axis. **c**, The enhanced mass spectrometer signals were time averaged during the interval of the 1:1 $N_2$:$H_2$ mixture to estimate the relative chemical reactivity. a.u., arbitrary units.

highest activity shown by any surface at any temperature (see Methods for further details). The reaction rate increases with increasing temperature and is higher for the stepped Fe(210) than the flat Fe(110) surface, in agreement with previous high-pressure-reactor studies[9]. The highest rate is seen for the Ru(10$\bar{1}$3) surface, as expected based on polycrystalline studies showing that Ru has higher activity than Fe (ref. 21). The maximum rate for Ru is not at the highest temperature of 723 K, as for the Fe surfaces, but at 623 K, also in accordance with catalytic-reactor studies[22].

On exposure to pure $N_2$ gas at 150 mbar, the two Fe surfaces have a delayed but eventually rapid increase in the N1s intensity, showing bulk nitride formation (Fig. 2a,b). On the basis of the binding-energy position of the N1s peaks in the spectra, this corresponds to the formation of γ′-nitride and ε-nitride plus some small amount of chemisorbed N atoms on the bare Fe surface (see Extended Data Table 1). The nitride formation is more rapid on the Fe(210) surface, specifically the γ′-nitride, whereas on the Fe(110) surface, there is an equal amount of the two nitrides and slower growth. The thicknesses of the nitride layers are greater than ten monolayers; exact quantification depends on the reaction time, as the surface continues to evolve even after hours of observation (see Methods for details on monolayer calculations). We attribute the faster growth on the Fe(210) facet to the higher probability of $N_2$ dissociation on the stepped surface[23]. At temperatures below 523 K, no nitride formation is observed.

The Ru(10$\bar{1}$3) reacts completely differently. Almost instantaneously after $N_2$ exposure, the N1s intensity saturates and remains constant, corresponding to a coverage of 5% of a monolayer, and there is no bulk nitride formation at 623 K (Fig. 2c). The coverage is comparable with previous work, which predicts 17% of a monolayer at 500 K and a pressure of 100 mbar (ref. 23). The small amount of $N_2$ on the Ru surface indicates a much weaker N–metal interaction than on Fe, as expected from theoretical predictions[16]. The two components are at 397.4 eV and 397.9 eV, and we tentatively assign these to N adsorbed on terraces and steps, respectively (Extended Data Fig. 1). It is interesting that a

weak, broad feature is seen at approximately 399–400 eV, with a binding energy consistent with adsorbed $N_2$ (ref. 24); see Extended Data Fig. 1.

When the pure $N_2$ gas is replaced by 1:1 $N_2$:$H_2$ at 300 mbar, a marked change on the two Fe surfaces occurs within the first spectral sweep (90 s), shown at the bottom of Fig. 2a,b. The nitrides instantaneously disappear and only a small amount of adsorbed N atoms with a coverage of 2% of a monolayer on Fe(110) and 5% on Fe(210) remains. At the same time as the gas mixture is introduced, $NH_3$ is detected by the mass spectrometer. The rapid removal of the nitrides shows the strong reduction ability of the $H_2$. The slow growth of nitrides (10–15 min) compared with the fast reduction (<100 ms) shows the difference in rates of $N_2$ and $H_2$ dissociation. The adsorbed N atom coverage is also substantially lowered on the Ru(10$\bar{1}$3) surface following the introduction of the 1:1 $N_2$:$H_2$ mixture at 300 mbar and decreases from 5% to <0.05% of a monolayer as $NH_3$ is produced.

Next, we address the question of oxides potentially not being reduced on Fe under operando conditions owing to trace contaminations of water or $CO_2$ in the gas phase[5]. Iron is known to oxidize in trace amounts of water or $CO_2$ at room temperature, yet iron oxide is not readily reduced below 500 K and, as a result, even under pure hydrogen, iron will oxidize with high flows (see Methods for a detailed description). Figure 3 shows data collected at 500 mbar, 1:3 $N_2$:$H_2$ and various temperatures. The Fe 2p$_{3/2}$ peaks in Fig. 3a from metallic iron at 706.5 eV and 707.4 eV are split owing to exchange interactions with the ferromagnetic valence electrons, and there is a broad Fe oxide peak at 710.8–709.8 eV, indicated by the grey rectangle. The Fe(110) sample is fully reduced as the temperature reaches 523 K at 500 mbar and the Fe(210) surface requires a higher temperature of 573 K, as seen in Fig. 3b. Fe(210) needs a higher temperature because of the stronger binding of oxygen on a stepped surface. Ru is metallic at all conditions. All surfaces are in a metallic state during the Haber–Bosch process, as expected because of the high concentration of adsorbed hydrogen (Fig. 3c). Note that these measurements were gathered simultaneously with the data in Fig. 4.

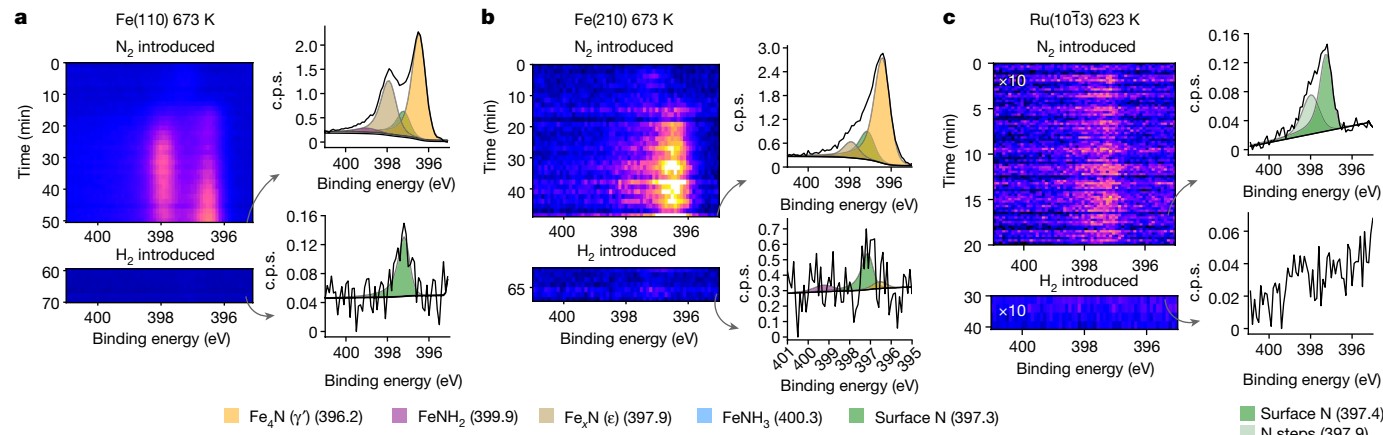

**Fig. 2 | Nitride formation and depletion.** The formation and depletion of nitride on the surface of each catalyst are shown as a function of time. At the top, the $N_2$ gas is introduced with a total pressure of 150 mbar and spectral collection begins. Then, after the nitride begins to stabilize, $H_2$ gas is introduced immediately in a 1:1 ratio with $N_2$ with a total pressure of 300 mbar, reducing the surface within the frame of the detector. Next to each time series are example spectra normalized to the background, with a grey arrow showing the frame it represents. **a**, The data for 673 K over Fe(110). **b**, The data for 673 K over Fe(210). **c**, The data for 623 K over Ru(10$\bar{1}$3). For Ru, the spectra shown are the summation of the entire time series. Note the difference in y-axis scale in the spectral figures.

The adsorbed nitrogen species can be measured operando as $NH_3$ is produced. First, focusing on the two Fe single-crystal surfaces (Fig. 4a,b), we observe only adsorbed N atoms on the surface at a binding energy of 397.4 eV, consistent with previous surface-science vacuum experiments once the recoil effect of the emitted atoms is considered (see Extended Data Table 1). Adsorbed molecular $N_2$ could not be detected and would have been observed at 399.0, 401.2 or 405.9 eV (Extended Data Table 1), depending on the adsorption site and bonding type. The coverage of adsorbed N is 1.3% at 200 mbar and 0.6% at 500 mbar on the Fe(110) surface and increases on the Fe(210) surface to 5.0% and 1.5%, respectively. The higher coverage on the stepped surface is related to availability and stronger bonding of undercoordinated sites[16]. What is most surprising is that the coverage is not increasing at higher pressures; on the contrary, the coverage decreases slightly with increased pressure. Inspecting the N1s spectra in Fig. 4d, measured at 1 bar and 673 K, the peak is barely distinguishable from the noise, implying an even lower coverage. It would be tempting to expect an increase in N coverage with increasing pressure because the imping-ing rate of $N_2$ molecules increases, but obviously also does the rate of H adsorption. Although we cannot determine the H coverage with XPS, our data suggest that the hydrogenation ability of the surface increases with the total pressure; this would explain a more efficient further reaction of the adsorbed N atoms. Extrapolating to much

higher pressures, we predict that the Fe surface is an almost pristine metal under realistic conditions. The fact that no amines (NH or $NH_2$) or $NH_3$ are observed at the reaction temperature of 673 K indicates that the rate-limiting step after $N_2$ dissociation is the hydrogenation of adsorbed N, and the rates of the other hydrogenation steps of NH and $NH_2$ as well as $NH_3$ desorption are much faster. At high temperatures, the Ru surface (Fig. 4g) has adsorbed N at 397.4 eV and the adsorb-ate coverage is almost negligible, with <0.1% of a monolayer of both NH and $NH_2$ species, independent of pressure within the noise limit. Here the surface is almost entirely clean of any species at conditions of high reaction rate.

At 523 K, for which the reaction proceeds very slowly, the population of the adsorbates changes. There is a slight increase of the adsorbed N on Fe(110) at 500 mbar to 2.3% of a monolayer (Fig. 4d). The Fe(210) surface shows large differences compared with the higher-temperature spectra (Fig. 4e). Further peaks at 398.0 eV, 398.9 eV and 400.2 eV formed, corresponding to NH, $NH_2$ and $NH_3$, as determined by previ-ous XPS vacuum studies[9,25,26] and calculated relative peak positions (Extended Data Table 1). Note that the peak at 399 eV is not related to adsorbed $N_2$ because ex situ XPS studies observed the peak when the Fe catalyst was cooled down to room temperature in the reaction mixture and moved to a vacuum, in which all molecular $N_2$ would desorb. We observe a relatively high coverage of $NH_2$ (24.8%), adsorbed N (4.3%),

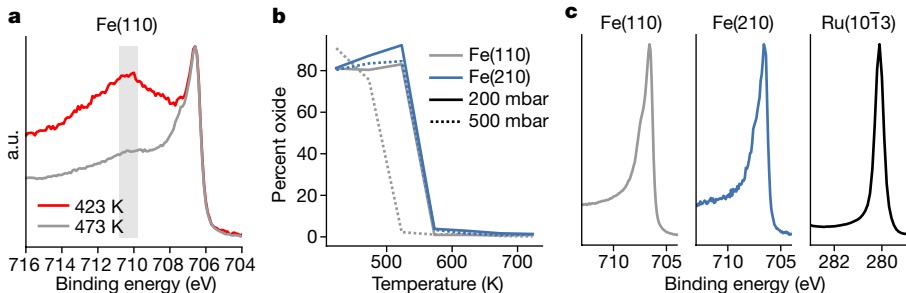

**Fig. 3 | Oxides and metal.** Owing to trace contaminations in the gases, the surfaces can form oxides. **a**, Two cases in which a thick oxide forms at low temperatures and 500 mbar in a 1:3 $N_2$:$H_2$ gas mixture, but the oxide thins and disappears as the temperature increases. The grey rectangle shows the region in which iron oxide peaks are present. **b**, The ratio of oxide to metal as a function of pressure and temperature for the Fe catalysts. The Fe(110) is grey, whereas the Fe(210) is blue. The solid line shows the lower-pressure data at 200 mbar, whereas the dashed line is the higher-pressure data at 500 mbar; at no point was the Ru catalyst oxidized. **c**, Example spectra of the metal peaks during $NH_3$ formation at 623 K, showing a singular metallic peak for all catalysts. a.u., arbitrary units.

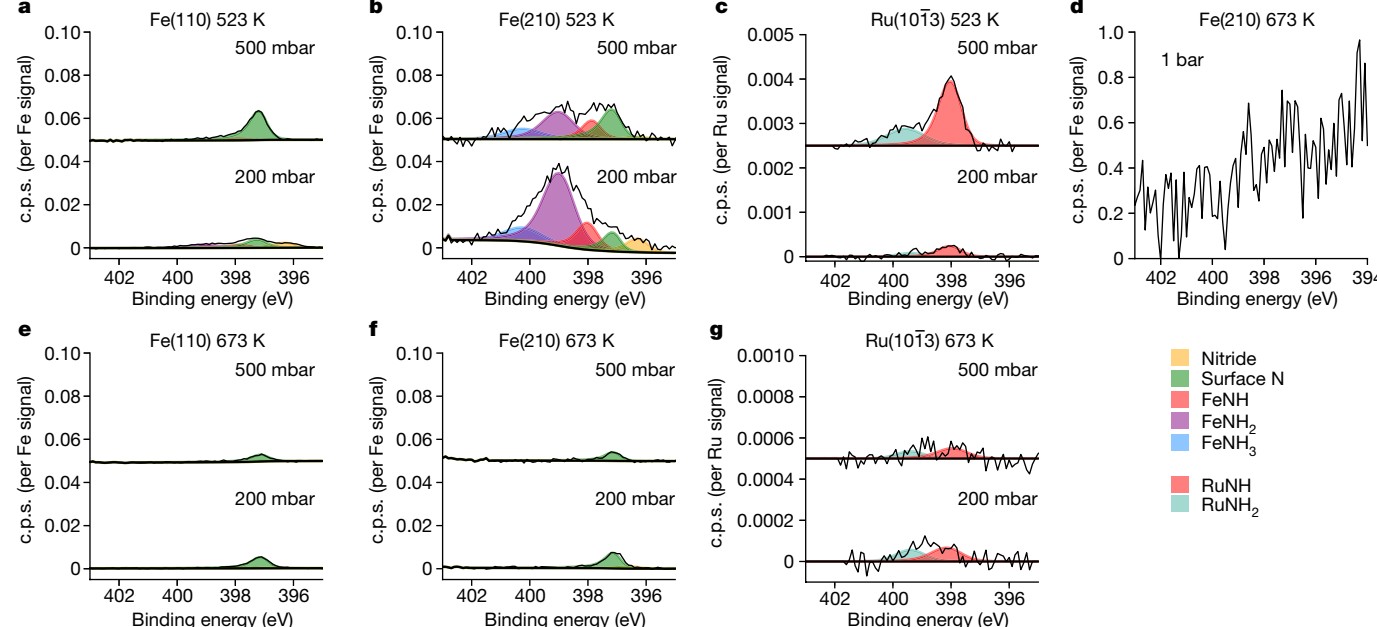

**Fig. 4 | Effects on adsorbates of temperature and pressure.** The steady-state population of the N species on the surface is shown for each catalyst at 200 mbar and 500 mbar at 523 K and 673 K in a 1:3 $N_2$:$H_2$ gas mixture. Each set of spectra is normalized and corrected for the cross-section of the corresponding metal substrate. **a**–**c**, The data over Fe(110), Fe(210) and Ru(10$\bar{1}$3) at 523 K, respectively. **d**, The data over Fe(210) at 673 K and at 1 bar. **e**–**g**, The data over Fe(110), Fe(210) and Ru(10$\bar{1}$3) at 673 K, respectively. Note the change in scale owing to the Ru data in **c** and **g**; nitrogen coverage of N species on the Ru surface is incredibly low.

NH (6.7%) and NH₃ (5.2%) at 200 mbar. There is a slight pressure dependence, for which—in particular—the NH₂ decreases to 9.3%. Clearly, there exist conditions in which the adsorbed N and NH$_x$ species are strongly adsorbed on step sites owing to a substantially lower hydrogenation rate. Decreasing the temperature further to 423 K, adsorbed NH$_x$ and NH₃ become visible on the Fe(110) surface. These trends are seen across 423 to 623 K (Extended Data Fig. 2).

On Ru at 523 K at 500 mbar (Fig. 4c), we still see very low coverages, although the coverage of adsorbed N at steps has increased to 0.5%, as well as adsorbed NH₂ to 0.1% and adsorbed NH₃ to 0.1% at around 400 eV. The NH signal increases with pressure, but the nitrogen coverage quantification of these results is nearly within the margin of error. If there is an increase in coverage with pressure for Ru, it may indicate that the H₂–metal interaction for Ru is weaker than for Fe, possibly leading to higher coverages at operational pressures. The adsorbed N species is much more reactive on Ru than Fe, supporting previous theoretical predictions[16].

We can discriminate the various proposed hypotheses and put forward ideas consistent with the data on the chemical state of the catalysts and reaction mechanism in terms of rate-limiting steps. We have shown that nitride formation is far slower than nitride reduction and that the surface states are all metallic with low coverages of atomic nitrogen. There is no evidence for interstitial nitrogen, oxides or high coverage of any species of nitrogen, especially over the most active catalysts. It is interesting to compare the hydrogenation reactions of CO and N₂, which are isoelectronic molecules. In the case of the Fischer–Tropsch reaction on Fe(110), a thick carbide is formed[15], whereas in the Haber–Bosch process, on the same surface, only a pristine metallic phase is generated. Clearly, the difference in the bond breaking of the CO molecule with respect to N₂ and the strength of the adsorbed C and N play an essential role.

The different reaction steps in NH₃ synthesis have been proposed as the following[10]:

$$N_2(g) + \theta^* \to N_2^* \tag{1a}$$

$$N_2^* + \theta^* \to 2N^* \tag{1b}$$

$$H_2(g) + \theta^* \to 2H^* \tag{2}$$

$$N^* + H^* \to NH^* \tag{3}$$

$$NH^* + H^* \to NH_2^* \tag{4}$$

$$NH_2^* + H^* \to NH_3^* \tag{5}$$

$$NH_3^* \to NH_3(g) + \theta^* \tag{6}$$

in which * means surface species and $\theta^*$ indicate empty sites available for bonding.

The simplest case is the Ru(10$\bar{1}$3) surface, for which we can directly explain that steps 3–6 are extremely rapid with no build-up of intermediates, pointing to 1 and 2 as the rate-limiting steps. We observe that the population of adsorbed N₂ is extremely low at high temperatures. The adsorbed molecular state is indeed observed at the low reaction temperature of 523 K, at which its dissociation limits the reaction. We conclude that the rate-limiting step of NH₃ production is the dissociation of the adsorbed N₂, fully in line with theoretical estimations[12]. Even at low temperatures, the surface is mostly adsorbate free, with little adsorbed NH$_x$ seen, because of the strong bonding to step sites in comparison with terrace atoms[23]. Although we have not observed definitive pressure dependence in the population of adsorbed N, it is plausible that the step sites will become more populated but are expected to remain well below a monolayer.

On Fe it is well established that the rate-limited steps is the molecular dissociation[7–9], supported by the correlation between the NH₃ production rate and the N₂ dissociative sticking coefficient for the different single-crystal surface facets[9,27]. However, the results here show that, at all temperatures, a factor of around 100 times higher

population of adsorbates is observed in comparison with the stepped Ru surface and we can no longer postulate that the reaction proceeds with a high rate after the molecular dissociative steps. Furthermore, there are no signs of molecularly adsorbed $N_2$ even at the lowest temperatures, indicative of a much higher rate of step 1b. Above 573 K, we observe adsorbed N that is more populated on the stepped crystal, indicating that the hydrogenation step 3 also partly controls the rate[12].

The coverage of N species on the Fe surfaces decreases with increasing total pressure at a constant $N_2$:$H_2$ ratio, implying that the $N_2$ dissociation step is slower than the hydrogenation step[10]. Most likely, the coverage of adsorbed H increases with pressure, resulting in faster hydrogenation. Because the coverage of $H_2$ at the reaction temperatures is expected to be low, we can assume that there is no inhibition of $N_2$ dissociation caused by the adsorbed hydrogen[27].

The population of intermediates shows that, as the reaction temperature lowers, the rate-limiting step switches to become hydrogenation of N, NH and $NH_2$ as well as $NH_3$ desorption (steps 3–6), demonstrating differences in the bonding at different high and low coordinated Fe sites. This agrees with earlier observations of the activation energy for hydrogenation being much higher than for $N_2$ dissociation[10] and the difference in the barriers of these two steps thus becoming prominent at low temperatures: although the $N_2$ dissociation rate at high temperatures is low owing to a low sticking coefficient that limits $N_2$ adsorption[10], we see a large population of amines $NH_x$ and $NH_3$ on Fe at low temperatures. This trend, not seen with Ru, points to the hydrogenation steps affecting the overall rate on Fe. At higher pressures at which more $N_2$ is converted and the $NH_3$ content is higher, the back reaction may become important. Indeed, for Ru, it has been theoretically predicted that the coverage of nitrogen species may become substantially higher[28].

In closing, we note that, although concerns over the environmental impact of ammonia synthesis have spurred interest in low-pressure alternatives and these might indeed be feasible[29], the Haber–Bosch process looks set to remain the primary method of ammonia production for many years to come. A better understanding of the mechanism at play might help to further improve the efficiency and, thereby, lower the environmental impact of this important industrial process. We anticipate that our approach to operando studies will contribute to this endeavour, by making it possible to explore the surface chemistry associated with ammonia formation in the presence of promotors and by making it possible, once measurements at higher pressures and with a higher $NH_3$ content are feasible, to explore the impact of the ammonia decomposition back reaction.

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

# Methods

## Ambient pressure XPS

All XPS data are collected with total external reflection X-rays and normalized to core levels of the substrate. The emitted photoelectrons and gases pass into the spectrometer through orifices in the front cone to be detected in a hemispherical analyser. The overall resolution in the measurement was 0.2 eV, all spectra were normalized with respect to the Fe $2p_{3/2}$ or Ru $3d_{5/2}$ core levels unless stated otherwise and spectra are presented in counts per second (c.p.s.).

POLARIS is an ambient pressure X-ray photoelectron spectrometer that operates with several key differences from typical ambient pressure XPS systems. The sample is approached to 30 µm from a set of roughly 20-µm-diameter apertures that lead to the analyser; the X-rays used are in the tender range 4.6 keV for all data collected. Most importantly, though, is that the gas is delivered through the front cone directly to the sample, making a virtual pressure cell in which only the sample and aperture to the analyser are pressurized. To achieve surface sensitivity, grazing incidence X-rays are used within the total external reflection range 0.3° for iron and 0.45° for ruthenium. This geometry allows for surface sensitivity despite high kinetic energy electron detection; the probe depths are 15.5 Å and 12.6 Å for Fe and Ru, respectively[30]. The electron spectrometer is a HiPP-2 hemispherical analyser manufactured by Scienta Omicron; see ref. 13 for more details. The single-crystal samples (Surface Preparation Laboratory, 99.99% purity) are mounted in a steel sample holder and heated from the back side with a resistive heater. The temperature of the sample is measured with a type C thermocouple pressed between the sample and the heater. The separation between the sample and the apertures is held constant by PID feedback based on the pressure over the sample[31]. A Si(311) double-crystal monochromator was used, yielding a photon-energy bandwidth of approximately 130 meV, a 0.8 mm curved entry slit and 100 pass energy was used in the electron analyser.

Extended Data Fig. 1a shows an example of an N1s spectrum of 1:3 $N_2$:$H_2$ gases at 200 mbar and the sample at 423 K, indicating adsorbed N atoms on the surface. The measurements were conducted at a photon flux at which no detectable X-ray-beam-induced changes could be seen. Individual spectra were gathered for 30 to 300 min with no decreeable spectral changes when hydrogen was present in the gas phase. Extended Data Fig. 3 shows an example time interval of 2 h over Fe(210) at 423 K and 500 mbar in 1:3 $N_2$:$H_2$ gas mixture. Extended Data Fig. 3a,b shows the data over this time for mass fragments 15, 16, 17 and 18, with and without processing. Extended Data Fig. 3c shows the XPS spectra evolution with time and Extended Data Fig. 3d shows the time-averaged results. From these, it is clear that the only change observed with time is the decrease in water signal owing to the slow improvement of vacuum conditions under constant hydrogen conditions.

## XPS data processing

All presented spectra are scaled by the number of sweeps and dwell time per data point. Further scaling is done based on the relative cross-section of the materials, as mentioned in the main text. To fit the spectra, CasaXPS was used with linear or Shirley backgrounds as needed. Peaks were fitted with modified Voigt function (LA) line shapes, which allows for asymmetry. Asymmetry was tuned for each component.

## Sample preparation

Sample cleaning was performed by ion sputtering with 3 keV $Ar^+$ for iron and 1 keV for ruthenium. The samples were annealed to 900 K for Fe and 1,100 K for Ru. Chemical cleaning was performed as needed by exposing the sample to either hydrogen or oxygen at elevated temperatures to remove oxygen or carbon, respectively. Small contaminations of sulfur and silicon were present, but the atomic composition was maintained at or below 1%.

## Coverage

To model the coverage of the surfaces, the $Fe_4N$ and RuN nitrides were used as the physical representation of the surface species. Although not a perfect model, reference data of commercial nitrides verify that the surface constituents are similar in atomic bonding. We used the method previously established[30] and typical XPS coverage formulation[32] to calculate the coverages. Elemental cross-section data was taken from ref. 33.

To calculate the probe depth, the X-ray and electron mean free path need to be combined; this is done by calculating the X-ray field in the material[34] at a given angle and using the TTP2M electron mean free path[35] to determine the electron-signal intensity as a function of depth within the sample. Then the integral is evaluated over all depths to determine the effective probe depth. Once the probe depth is determined, the coverage is then calculated on the basis of the ratio of substrate to surface species intensity-weighted by the cross-sections and atomic densities[32].

## Mass spectrometry

To determine the amount of $NH_3$ formed by the catalyst, a differentially pumped mass spectrometer (Hiden HAL/3F RC 301 PIC system) was attached to the first differential pumping stage of the XPS analyser. By leaking a small amount of gas from the pumping stage to the mass spectrometer, the composition of the gas over the sample was determined. To ascertain the gas composition, mass fragments of all relevant peaks were monitored. Impurities in the $N_2$ and $H_2$ gases were predominantly $H_2O$ and $CO_2$. These contributions to the $NH_3$ fragments were subtracted on the basis of the measured ratio of pure gas to contaminate. Owing to the marked overlap of water and $NH_3$ ionization patterns, $m/z = 15$ and 16 were used as the markers for $NH_3$. Further smoothing is done with a third-order Savitzky–Golay filter over a window of 1 s. The result of this analysis is shown in Fig. 1. As the mass spectrometer is highly sensitive, there is signal before any experiment from the chamber at all masses, including masses 15 and 16, most likely because of hydrocarbons. With the high-flow conditions required to establish the pressure for the XPS measurements, the amount of ammonia in the gas stream into the mass spectrometer is small and the signal becomes noisy. Therefore, to make a more accurate measurement of the $NH_3$ production, time integration was done between the background level in pure $N_2$ and that of pure $H_2$. The background was subtracted from the time integration during ammonia production. Extended Data Fig. 4a shows an example mass spectrometer time trace in which there is negligible $NH_3$ production, showing how the background change with gas flow is within the noise of the measurement and therefore requires time integration. Owing to the specific design of the high-flow virtual cell, unwanted gas molecules originating from reactions of the sample holder or heater cannot reach the single-crystal surface area that is examined by the opening into the electron spectrometer. Thereby, all measurement conditions are constant. The increase in ammonia production at higher temperatures is as expected according to refs. 9,21,22, providing further confidence that ammonia is produced.

The relative chemical activity (RCA) was calculated using the following equation. Time-averaged $NH_3$% was calculated from the amount of signal from ammonia as described above per total signal from the mass spectrometer. Volume ($V$) of gas is the total volume of gas used during the measurement, pressure ($P$) over the sample, temperature ($T$) of the sample, the gas constant ($R$), time is the duration of the time when ammonia could have been produced, $A_n$ is Avogadro's number and sites is the number of active sites under the high-pressure area. Finally, the highest activity on any surface is a normalization to the maximum of any surface. The normalization is to account for systematic errors, such as the fact that most of the volume of gas used does not pass over the sample or the fact that not all sites in the high-pressure region under the front cone would be examined by the mass spectrometer.

$$RCA = \frac{\overline{NH_3\%} \times V \times P}{sites \times time \times T \times R \times A_n} / \text{highest activity on any surface}$$

With an instrument exposed to many gases over the years, there are signals at all masses, including masses 15 and 16, before any ammonia-synthesis experiment is performed, owing to desorption from the chamber walls. This desorption in the first differential pumping stage most likely comes from hydrocarbons. In the mass spectrometer, it is possible for crosstalk between channels or other instrumental errors to affect the signal. This is particularly true when the signal is very near the noise level, as in the work presented herein. Extended Data Fig. 3 shows the masses 15, 16, 17 and 18. Mass 17, corresponding to ammonia, is strongly affected by water production from $H_2$ interaction with the chamber walls and the mass spectrometer itself, making quantitative analysis impossible. To decrease the possibility of the ammonia signal originating from instrumental errors, both masses 16 and 15 are included in the signal of ammonia. As discussed above, mass 17 is not included because of the large water signal. Extended Data Fig. 3a,b shows the effect of both processing and long acquisition times. The sample is Fe(210), 423 K, 500 mbar, 1:3 $N_2$:$H_2$ ratio. Here we can see that the atomic mass units of both 15 and 16 are constantly above the background signal from hydrocarbons or water. Meanwhile, mass 17 is not, owing to the strong overlap of OH and $NH_3$ masses. Note that the background signal removed at this point in the processing does not account for all of the background signals. As described above, to determine the relative chemical activity, the signal of ammonia (masses 15 and 16) above the background signal in either pure $N_2$ or pure $H_2$ is taken. That ammonia signal is then compared with the total signal in the mass spectrometer over the same time period. By this method, the plotted data do not remove all of the background signals, yet when the data are processed for relative chemical activity, the entirety of the background is removed.

The error of the relative chemical activity is estimated on the basis of the signal-to-noise ratio of the background of the ammonia signal. Part of the calculation is to subtract the background, seen in Fig. 1b, between times 15 and 23 min; the fluctuations in the background can have a notable effect on the calculation of ammonia content. To ascertain the estimated error, the 95% confidence interval of the noise average and standard deviation over the collected time were introduced as an error source in the equation for relative chemical activity. Because the background signal and noise are similar for all experiments, the estimated error introduced is also similar. The relative chemical activity is meant to be a semiquantitative description of the abundance of ammonia, only as a comparative description of these similar systems, and to demonstrate that the trends follow previous more absolute activity measurements. Extended Data Fig. 4a shows an example of when there is no ammonia production at the lowest temperature with the least active catalyst, Fe(110) at 523 K (300 mbar, 1:1 ratio). Here we can see that extremely small ammonia production occurs and this is most likely the background level. Extended Data Fig. 4b, by contrast, shows the same surface and experiment at the higher temperature of 673 K and clearly shows that ammonia production increases with increasing temperature.

## Beam effects

To determine the effect of the X-ray beam intensity on the observed species, two types of beam damage test were performed, attenuation and dark spectra, both carried out at 423 K, 500 mbar and a 3:1 mixture of $H_2$ and $N_2$. The dark spectra were performed by aligning the sample and gathering highly attenuated spectra, then closing off the light, cleaning the sample in hydrogen, then opening the shutter and obtaining a new spectrum. The result showed no change owing to the amount of time the sample was exposed to X-rays. The second test was done by gradually increasing the intensity of the beam to determine whether any beam damage would accumulate; here no change in the spectra once normalized to the attenuation factor was observed.

Nitrogen spectra gathered outside reaction conditions (that is, without hydrogen), such as those shown in Fig. 2, show not only chemical changes inherent to the reaction but also accumulation of beam-induced effects. The build-up of beam-induced nitride formation is slower than chemical activity but is not possible to fully avoid. For this reason, no attempt is made to quantify the formation rate of the various nitrides. The main finding of the paper is the distinct lack of nitrogen on the surface during the reaction and the slow nitride formation compared with fast reduction caused by hydrogen. The beam-induced nitride formation only serves to increase nitrogen formation. Therefore, the beam effects do not alter any conclusions of the activity of $N_2$ compared with $H_2$ but rather strengthen the finding.

To determine whether the beam has any effect on the mass spectrometer findings, two experiments of gas switching (in which the sample was exposed to pure $N_2$ then 1:1 $N_2$:$H_2$ then pure $H_2$) were performed with X-ray light and without. Extended Data Fig. 4b,c shows the mass fragments of all relevant species for these experiments. For these data, the sample was Fe(110) at 673 K. Extended Data Fig. 4b shows the measurement with beam, whereas the measurement in Extended Data Fig. 4c was collected without beam. Although there are some minor differences in the ammonia signal between the two datasets, none of the changes are what would be expected from beam effects. It is expected that beam effects in the mass spectrometer would strictly cause an increase or decrease in ammonia signal. The change in the relative chemical activity between the test done with and without beam is approximately 2% and well within the error of the experiment. From this, it is clear that the X-ray beam does not have any effect on the mass spectrometer findings.

Extended Data Fig. 5a shows the effect of photon flux on the nitrogen content. Note that the line of best fit shown in grey has a forced intercept to zero. Extended Data Fig. 5b shows the effect of flux on nitrogen speciation, showing no change to the components of the N1s spectra. These spectra were gathered at 500 mbar in a 1:3 gas ratio at 423 K over Ru, the conditions expected to be the most sensitive to beam damage. Equivalent tests were performed for all catalysts.

## Trace contaminations

Although highly pure gas (5N for nitrogen and hydrogen) was used with in-line chemical purifiers ($N_2$, model no. MC45-804; $H_2$ gas, model no. MC45-904, SAES Group), trace impurities are still present; on the basis of the mass spectrometer data, approximately 6 ppm of water and 3 ppm of $CO_2$. Owing to the high flows used, these small contaminants can react and build on the surface instantly. Furthermore, molecules will readily react with iron to form iron oxides; the same is not true for ruthenium. As a result, the iron surface at low temperatures will form a partial oxide but, as the temperature increases, the reduction by hydrogen outpaces the oxidation of the contaminants, yielding a metallic surface at relevant conditions.

## Nitrogen species binding energies

Extended Data Table 1 presents previously published and computed binding energies for various amine and nitrogen species over iron and ruthenium. Note that, for comparison with studies at low photon energy, the recoil effect, in which the energy of the electron causes the nucleus to recoil, thereby decreasing the kinetic energy of the emitted electrons, needs to be considered[36].

## Data availability

Experimental data were generated at the PETRA III facility at the DESY Research Centre of the Helmholtz Association. Raw datasets are available from the corresponding authors on reasonable request.

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

**Acknowledgements** This research was financed by the Swedish Research Council (Vetenskapsrådet, VR) under award numbers VR 2013-8823 and 2017-00559, as well as the Knut and Alice Wallenberg Foundation under award number KAW 2016.0042. We acknowledge Deutsches Elektronen-Synchrotron DESY (Hamburg, Germany), a member of the Helmholtz Association (HGF), for the provision of experimental facilities. Parts of this research were carried out at PETRA III using beamline P22. Beamtime was allocated for proposals I-20200291 EC, I-20200292 EC and II-20211048 EC. J.K.M. is grateful for financial support from VILLUM FONDEN (research grant 41388).

**Author contributions** A.N. and C.M.G. proposed the study. C.S. developed the beamline. C.M.G., A.N., D.D. and P.A. developed the experimental set-up. C.M.G., P.L., D.D., M.S., P.A., F.G.-M., S.K., B.D., J.K.M. and R.R. performed the experiments. C.M.G. analysed the data. G.L.S.R. performed the binding-energy calculations. A.N., C.M.G. and B.D. wrote the manuscript.

**Funding** Open access funding provided by Stockholm University.

**Competing interests** The authors declare no competing interests.

**Additional information**
**Correspondence and requests for materials** should be addressed to Christopher M. Goodwin or Anders Nilsson.

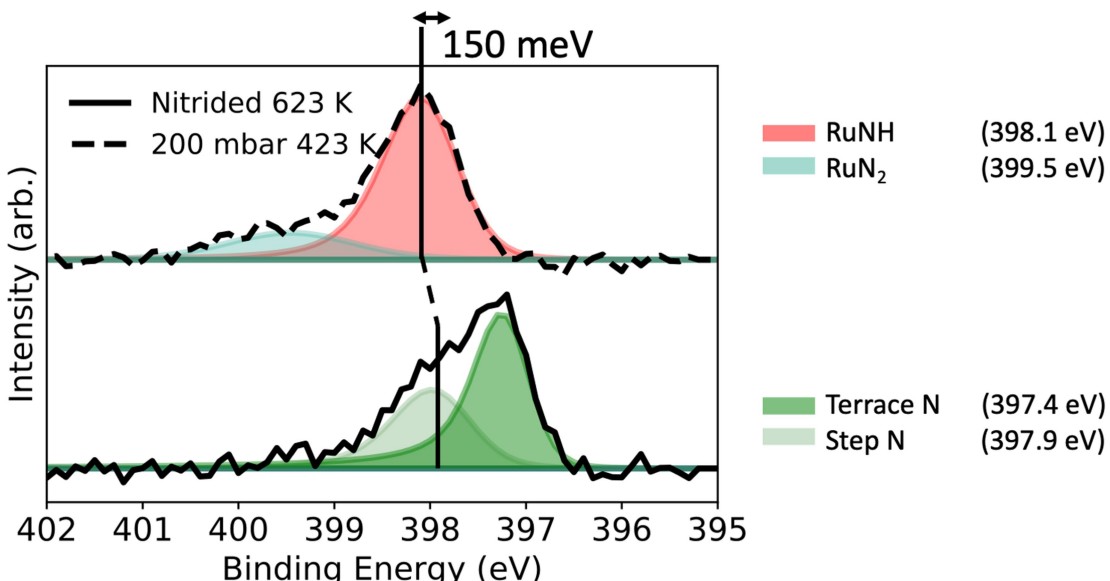

**Extended Data Fig. 1 | Comparison of N1s over ruthenium under pure N₂ and low-pressure condition.** The spectra of N1s on Ru(10$\bar{1}$3) under 200 mbar 1:3 N₂:H₂ gas mixture at 423 K and the thickest nitride film made on the same surface at 623 K. The black lines show the offset between the NH species in red in the top spectra to N on terraces and light green in the bottom spectra cumulating to 150 meV.

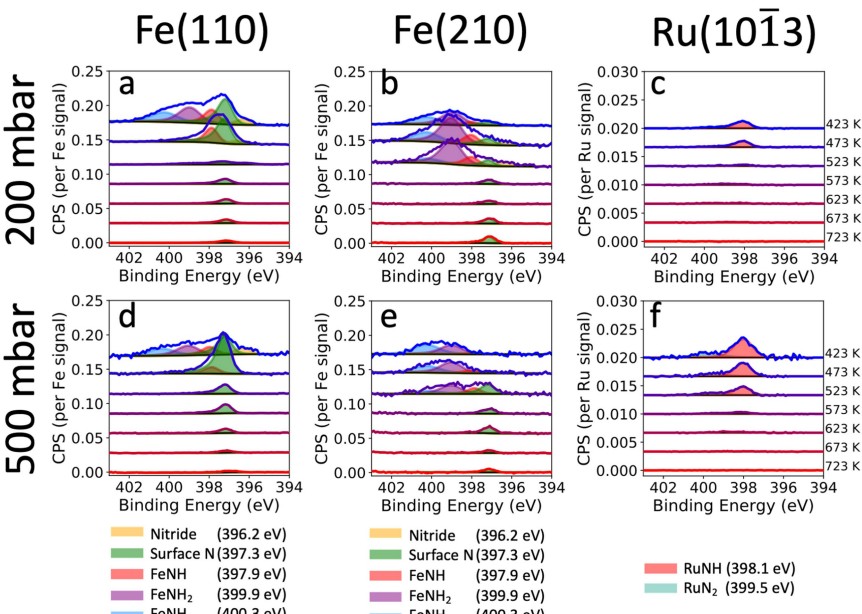

**Extended Data Fig. 2 | N1s data at every temperature and pressure. a–c**, 200 mbar 1:3 $N_2$:$H_2$ nitrogen spectra over Fe(110), Fe(210) and Ru(10$\bar{1}$3), respectively. **d–f**, 500 mbar 1:3 $N_2$:$H_2$ nitrogen spectra over Fe(110), Fe(210) and Ru(10$\bar{1}$3), respectively. Temperature increases from top to bottom, from 423 K to 723 K.

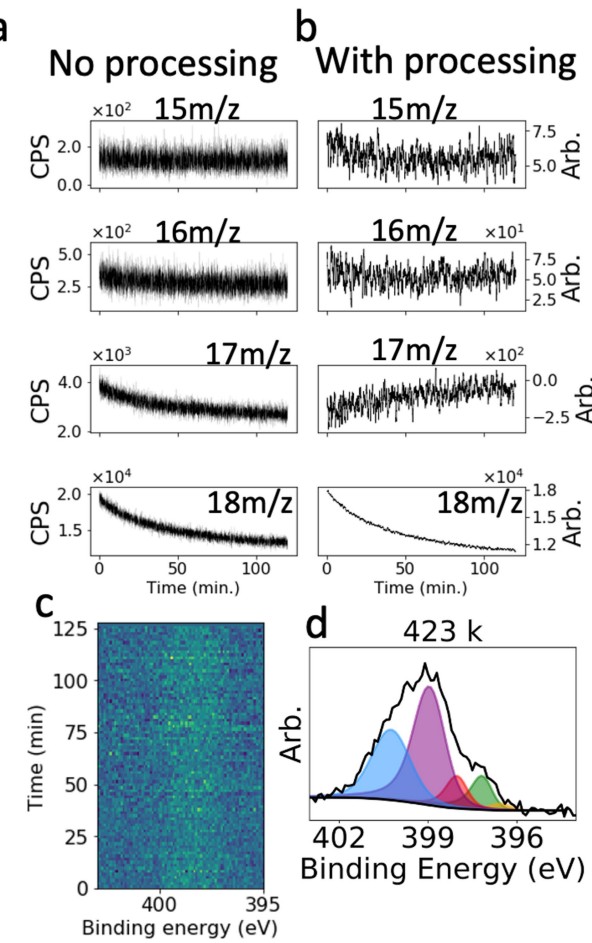

**Extended Data Fig. 3 | Stability of long acquisition times. a**, The set of mass spectrometer data before any processing gathered over a 2-h window. The trend in masses 16, 17 and 18 are because of the slow improvement of the vacuum conditions. **b**, The same data as in **a** with processing as described in Methods. **c**, XPS spectra gathered simultaneously normalized to background signal. **d**, The sum of the spectra with the fitted peaks shown according to the colour scheme used throughout the paper.

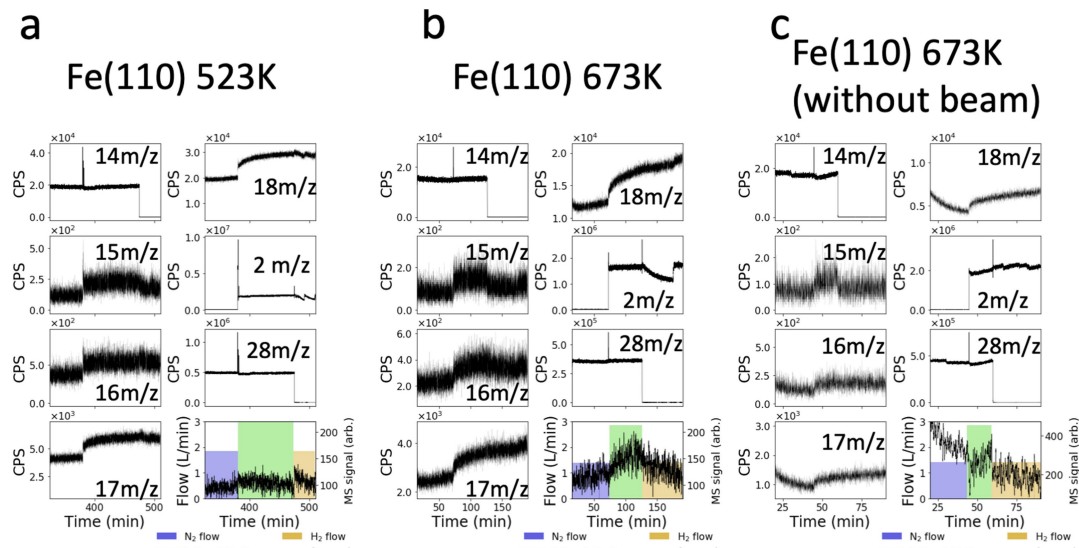

**Extended Data Fig. 4 | Mass spectrometer time trace with and without ammonia production. a**, The equivalent experiment as for the data shown in Fig. 1b but at 523 K over Fe(110) with the complete set of mass fragments shown before any data processing, as well as the ammonia signal with all of the data processing. **b**, The equivalent experiment but at 673 K over Fe(110). Note here that the timescale is far longer than in Fig. 1b to ensure sufficient statistics,

ensuring that the relative reactivity is representative of the lack of ammonia production. Note that, in **a**, there is a glitch in the mass spectrometer background as a response to the removal of $N_2$, but it is clear that mass 15 is decreasing in pure $H_2$ after the switchover. **c**, The equivalent experiment as **b** but performed without any X-rays on the sample.

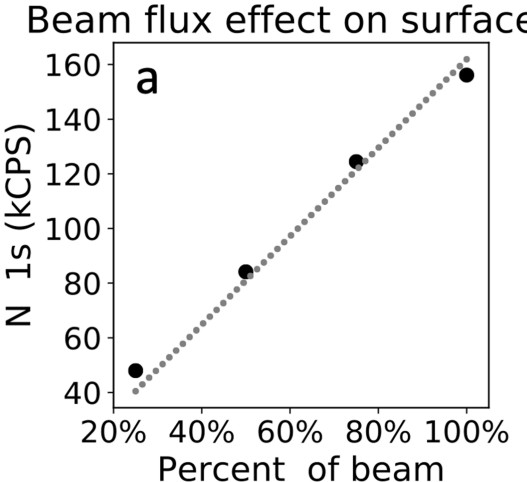
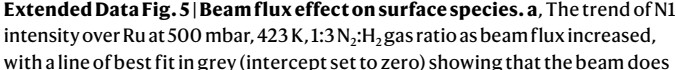
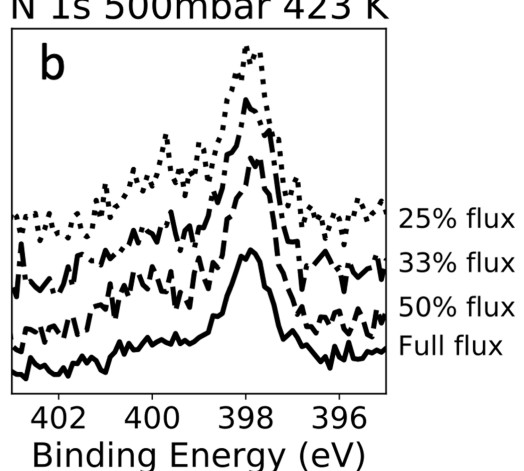

**Extended Data Fig. 5 | Beam flux effect on surface species. a**, The trend of N1s intensity over Ru at 500 mbar, 423 K, 1:3 N₂:H₂ gas ratio as beam flux increased, with a line of best fit in grey (intercept set to zero) showing that the beam does not have an appreciable effect on total coverage. **b**, The same experiment showing that the N1s spectra gathered sequentially from lowest flux to full flux, showing no change in speciation with beam flux.

**Extended Data Table 1 | Nitrogen 1s binding energies**

| Chemical state | | Iron | Calculated iron (110) | Ruthenium |
|---|---|---|---|---|
| Nitride | $\gamma$'−nitride | 396.7[37] | 397.9 | 398.4[38] |
| | $\varepsilon$−nitride | 397.9[39] | 397.9 | |
| Surface N | | 397.3[8,17,25,27] | 397.4* | 397.4[40] |
| $N_2$ Terrace | | 405[24,25] | 399.3 | 399.9[†23] |
| NH | | 398[41] | 398.1 | 397.9[42] |
| $NH_2$ | | 399[8,17,24,27] | 399 | 398.5[40,42] |
| $NH_3$ | | 400[8,17,24,27] | 400.7 | 399.7[40] |

*Calibration binding energy.
†On the basis of the average position of both nitrogen atoms in $N_2$. Refs. 8,17,23–25,27,37–42.