## [Peer Review File · Nature]

Manuscript Title: Operando Probing of the Surface Chemistry During the Haber-Bosch Process

Reviewer Comments & Author Rebuttals

Reviewer Reports on the Initial Version:

----- Referees' comments -----

Referee #1 (Remarks to the Author):

This is a great paper showing experimentally that the state of the surface (and bulk) of Fe and Ru catalysts are metallic and that N₂ dissociation is rate limiting at usual synthesis temperatures, while N hydrogenation may become rate limiting at lower temperatures. All of this confirms the picture that has emerged through extensive modeling based on DFT calculations in the Norskov group over decades. Showing experimentally that it is so is obviously extremely important. And this did only become possible recently through the extremely advanced operando, high temperature XPS developed in the Nilsson group. Ammonia synthesis has long been the testbed for our understanding of heterogeneous catalysis, and for that reason alone, this is a very important paper worthy of publication in Nature. It simply closes a lot of the speculation in the field.

I have a suggestion or two. The discussion of surface coverages is based on rates of different hydrogenation elementary steps. In fact, at the higher temperatures all steps beyond N₂ dissociation are in equilibrium and hence the coverages are related to the thermodynamics and the ammonia partial pressure. All experiments are at low conversion, hence the low coverages. At the lowest temperatures N hydrogenation is shown to be rate limiting. A nice prediction of that and a discussion of the reasons why can be found in Journal of Catalysis 220, 2003, Pages 273-279.

Referee #2 (Remarks to the Author):

In the article "Operando Probing of the Surface Chemistry During the Haber-Bosch Process", by Goodwin et al., the authors use ambient pressure X-ray photoelectron spectroscopy to track the surface species and oxidation state of single crystal model catalysts for the formation of ammonia from nitrogen and hydrogen. This is a very important process in the industry, which is not fully understood, and studies like the one presented here are extremely important. The tools that the authors use are cutting-edge and appropriate for this study. However, in order to properly assess the validity of the results there are a few issues that need to be addressed. I would be happy to review this paper again after these issues have been taken care of. The paper has lots of conclusions from data that are not explicitly shown. Please include this data as xy files (or another format) in the supporting information. Others should be able to access this data to verify the statements, rather than just trusting the authors' words.

1. Text in figure 1a is too small and impossible to read.
2. Figure 1b is not properly described. It looks like it is a gas flow and mass specs signal vs time plot, but that's not what it says in the caption.
3. Also in figure 1b, what are the units of the mass spec signal? Mass 15 is tracked for ammonia, but this is not explicitly said in the figure or caption. Why is there a signal before adding H₂? Signal increases from ~10 to 12 (in whatever units that is). The authors do not include the mass spec data for any other gases. It is common in mass spectrometer that there is some cross-talk between channels (especially for very small signals like this one), how do the authors know that this increase is not from cross talk with mass 2 from H₂? How does mass 28 from N₂ change when H₂ is added? Please include this data at least in the supporting information. This is critical, as some of the main conclusions of the paper are based on this data.
4. Figure 1c. Can error bars be included in any way? This calculates a TOF based on the mass spec data. It is not clear enough how this TOF is obtained. Please include the data used to obtain the TOF in the supporting information and an example of how the data is converted into TOF.
5. Given that these experiments use synchrotron light, beam effects have to be considered. The authors describe these studies in the paper, but this description is not detailed enough to understand how they did these tests, and none of the data from these tests is included. Please include the data to support your statements.
6. The authors state that nitrides do accumulate with time under the beam. Isn't this expected to affect the conclusions? or are these nitrides not expected to participate in the reaction? Does that mean that the beam facilitates the breaking of the NN bond?
7. For much of the data spectra are shown as a function of temperature and pressure. However, no time dependence is described. For how long was the sample exposed to these conditions? Is there any change with time on the surface?

Referee #3 (Remarks to the Author):

Review report on manuscript:

Operando Probing of the Surface Chemistry During the Haber-Bosch process

by Christopher M. Goodwin, Patrick Lömker, David Degerman, Mikhail Shipilin, Fernando Garcia-Martinez, Sergey Koroidov, Bernadette Davies, Jette Katja Mathiese, Raffel Rameshan, Gabriel Libanio Silva Rodrigues, Christoph Schulter, Peter Amann, Anders Nilsson

In this paper the authors use the unique POLARIS APXPS instrument to Fe(110), Fe(210), and Ru(10-13) model catalyst surfaces exposed to N₂ and N₂:H₂ mixtures at total pressures between 150 mbar – 1 bar. The data presented are quite nice and I enjoyed reading the paper. Correct labeling of figures and an occasionally more consistent discussion would have made the reading experience even better (see my detailed critics below).

The key finding of the work is that all 3 surfaces remain metallic at high temperatures of 673 K, with low surface coverage of adsorbates. Furthermore, the authors identify differences between stepped Fe and Ru surfaces and convincingly demonstrates that the rate-limiting step at 673 K is N₂ dissociation on Ru, while the hydrogenation step also is essential for the total rate on the stepped Fe surface. While the APXPS data mostly are convincing (see my detailed critics) below, I am more unsure whether the NH₃ formation are demonstrated convincingly with QMS on all studied surfaces (see also my detailed critics below).

The theme of the paper is related to recent article published articles in Science (ref. 27) and ACS catalysis (ref. 25) by the same group. In their previous work the authors studied CO₂ and CO hydrogenation over a Zn/ZnO/Cu(211) surface to mimic real CuZn-based catalyst used for methanol synthesis (ref. 27) and Fischer-Tropsch synthesis (ref. 25). Therefore, I would say that the novelty of POLARIS instrument already has been demonstrated and communicated. Thus, this cannot be used as an argument to publish in Nature.

I am not an expert in the literature of Haber-Bosch process, but as far as I can read different surface terminations have been suggested at operational conditions and also different rate determining steps have been suggested. The present article use for the first time APXPS to show that the studied surfaces at pressures close to one bar are metallic and that N₂ dissociation is the rate determining step on stepped Ru while the hydrogenation step is suggested to play a role also on Fe surfaces.

Whether the first APXPS studies performed close to 1 bar pressure or 2-3 orders of magnitude higher than conventional APXPS studies (but 2 orders of magnitude lower than real conditions) is sufficient to warrant publication in Nature is difficult for me judge. This is most likely better judged by experts in the Haber-Bosch process that have a complete and up-to-date overview of the current understanding of this process. Below follows my detailed critics which I hope the authors can use to improve their article.

If the editor decide that novelty is insufficient for publication in Nature I believe the article fits well in Nature catalysis or Nature communication.

Major points:

1. In general, I miss details for the QMS data.

It is a bit unclear from text how the results of figure 1c have been conducted. On p. 4, l. 107 – 109 the authors write. “The relative turnover frequencies at various temperatures and between single crystal catalysts were determined by calculating the percent NH₃ formed in the gas stream at a pressure of 300 mbar, shown in Fig. 1c.”. The figure caption of figure 1b mentions a gas composition of 1:1, but it is unclear from the text if all datapoints in figure 1c were conducted at 300 mbar total pressure and a 1:1 gas composition. After carefully reading the method section I understood that indeed a 1:1 gas composition was used. I encourage the authors to state this clearly in the text. The authors discuss “relative TOF in arb. units” a term I am not familiar with. As I understand the methods section the authors calculate this by first finding the relative gas composition of NH₃ and then dividing this by the number of sites on the catalyst surface.

I am not sure that the general readership of high impact journals will understand the thoughts behind this. I would suggest that the authors in the text or the method section mention that the partial pressure of NH₃ is proportional to the turnover frequency, i.e. the number of reaction turnovers per second and that they then normalized this to the atomic density of Fe or Ru atoms (assuming no reconstruction of the surface).

An alternative would be to state the relative conversion per site in figure 1c where 100% refer to reaching the mass transfer limit. This is easier to understand for general readers I think. Also one avoid “arb. units”.

I assume that the authors assumed that the reaction only takes place at the front sites of their crystals? No information about this is given in the text and if so the authors need to motivate why this can be done.

Finally, I assume that the authors did not calibrate the QMS with known gas compositions and miss a sentence about this in the method section if this were the case.

2. As the authors state themselves little NH₃ is produced. Therefore, one also needs to be very careful with the evidence for the NH₃ activity. On p. 4. L. 109 – 111 it is said that: “The reaction rate increases with increasing temperature and is higher for the stepped Fe(210) than the flat Fe(110) surface in agreement with previous Fe single crystal high pressure reactor studies”. Looking at figure 1c I honestly do not see much evidence for increased reaction rate on the Fe(110) surface. Maybe a little for the points at 400 dC and 450 dC, but it is very little and without any error bars, very few data points, and no information about the where the baseline is (i.e. activity at RT) it is difficult to evaluate.

I also miss a blind experiment on a sample holder without any crystal. Is there really zero NH₃ formation in this case at 450 dC? I would suspect this, since the sample holder most likely has a higher temperature than the single crystal surface due to the conductive heating system used. Without such a reference experiment it is difficult to be sure that the very low NH₃ activity does not come from other hot heated parts near the sample surface.

3. I am confused about the pressure and temperature used for the experiments shown in figure 2. In the figure caption to figure 2 it says that spectra collection begins once a pressure of 150 mbar is reached. In line 128-129 on p. 4 the text states that the samples are exposed to 250 mbar? Finally, the extended data for figure one states 200 mbar? I have simply no idea what gas mixture that was used for the data collection in figure 2a, 2b, and 2c. Please specify this so it is clear.

4. Why are dC used in figure 1c, while K is used in figure 2?

5. I miss information in the text why temperatures of 673 K is used for the iron surfaces, while 623 K is used for the Ru surface. The authors say in line 113, p. 4 that "The maximum rate for Ru is not at the highest temperature of 723 K, as for the Fe surfaces". I see that the highest rate for Ru is at 623 K, but line 113 seems to imply that the highest rate on Fe surfaces is observed at 723 K. Then one should have measured at 723 K for Fe, not at 673 K? Maybe the reason is that the rate at 673 K and 723 K is almost the same on Fe? If so the authors should explain this.

6. The signal to noise ratio of figure 2a and 2b looks quite different. In fact it does not look like figure 2b is normalized to the background. Some lines look quite intense while others look weak. The authors should correct this mistake or comment why the signal to noise ratio is so different in figure 2a and 2b. Also figure 2c looks like no calibration to the background has been performed.

7. In line 153-154 the authors write: "It is interesting that a weak broad 153 feature is seen at \approx 399-400 eV with a binding energy position consistent with adsorbed N₂." It took me quite some time to realize that the authors here refer to the extended figure 1. The sentence should be reformulated such that this is clear.

8. I like the paragraph in line 155-167 on p. 5 and I believe the authors presented solid evidence for slow N₂ dissociation and much faster H₂ dissociation and subsequent reaction at sub-bar pressures. This is good!

9. The paragraph in line 179-191 on p. 6 convincingly shows that both Fe and Ru surfaces are metallic at elevated temperatures. This is good. However, I had initially to spend some time to understand why other pressures and gas flows were used for figure 3 than in the discussion for figure 2. I therefore encourage the authors to briefly discuss the motivation behind the experiments in figure 3 better.

10. It looks like the panel labels in figure 4 is mixed up. In line 194-195, p. 6 it is written that: "Fig. 4abcd shows the N1s spectra of the surfaces at 673 K and different pressures with a 1:3 N₂:H₂ synthesis gas mixture". This must be a typo? Figure 4a-d shows data recorded at 523 K. Also later in line 216 p. 7 fig. 4c should be 4g I assume. Line 249, p. 7: "Fig. 4f..." should be "Fig. 4g...." I assume. The authors should carefully check that all labels are correct in particular for figure 4.

11. I do not understand the way the Ru data are discussed for figure 4. In the last paragraph on p. 6 the authors first focus on Fe spectra recorded at 673 K (discussion starts on line 195). Following this the authors briefly discuss similar data for Ru starting line 215 (even though they refer to the wrong figure. I assume fig 4c should be fig. 4g instead).

Subsequently, data recorded at 523 K are discussed. Again, Fe surfaces are discussed first (line 231, p. 7).

Finally, in line 249, p. 8 data for Ru are discussed, but only for 673 K? Where are the Ru data in figure 4c discussed? Also, there is a clear increase of the RuNH component with pressure.

12. I do not understand the sentence in line 312-314 on p. 9: "Since the coverage of H₂ at the reaction temperature is expected to be low, we can assume that there is no inhabitation of adsorbed hydrogen for the N₂ dissociation". Maybe the authors mean: "Since the coverage of H₂ at the reaction temperature is expected to be low, we can assume that there is no inhabitation due to/caused by adsorbed hydrogen for/on the N₂ dissociation"

13. I am afraid that the discussion/conclusion for Fe surfaces starting line 274, p. 8 and ending line 326, p. 9 is difficult to read for readers of high impact journals. In fact, I am not sure that I understand the take-home message myself. For example, line. 311-312 on p. 9 say that "...directly implies that the N₂ dissociation step is slower than the hydrogenation step of the adsorbed N (here the authors discuss the Fe)" while it a few lines above is written that: "we observe adsorbed N that are more populated on the more strongly bonded stepped crystal, and the hydrogenation step 3 also partly controls the rate". To me it is unclear what controls the rate on the Fe surfaces and the text reads like different things are said at different places. Another example is the discussion of figure 2 where it also is highlighted that H₂ is very quickly removes N-species from the surface.

14. In line 331 p. 10 the authors write: "With improved instrument sensitivity allowing high quality spectra to be measured in seconds or minutes (not as here many hours) will provide..." sounds like claim without any support. As I understand it the instrument sensitivity of POLARIS is limited by the incoming flux and the electron scattering in the gas phase. For such a statement the authors need to discuss that it is feasible to reach 60 or 3600 higher overall instrument sensitivity.

Minor parts:

15. "N1s" and all other core levels should be written with a space between the element and the core level ("N 1s")

16. Some of the figure text is way to small to be read on a standard printout of the article. This is for example the case for embedded text in figure 1a, many captions, etc.

Author Rebuttals to Initial Comments:

Reviewer: 1

First we would like to thank the reviewer for taking the time to read the manuscript and provide a critical review. The comments have helped us to develop a stronger and clearer manuscript.

His is a great paper showing experimentally that the state of the surface (and bulk) of Fe and Ru catalysts are metallic and that N₂ dissociation is rate limiting at usual synthesis temperatures, while N hydrogenation may become rate limiting at lower temperatures. All of this confirms the picture that has emerged through extensive modeling based on DFT calculations in the Norskov group over decades. Showing experimentally that it is so is obviously extremely important. And this did only become possible recently through the extremely advanced operando, high temperature XPS developed in the Nilsson group. Ammonia synthesis has long been the testbed for our understanding of heterogeneous catalysis, and for that reason alone, this is a very important paper worthy of publication in Nature. It simply closes a lot of the speculation in the field.

I have a suggestion or two. The discussion of surface coverages is based on rates of different hydrogenation elementary steps. In fact, at the higher temperatures all steps beyond N₂ dissociation are in equilibrium and hence the coverages are related to the thermodynamics and the ammonia partial pressure. All experiments are at low conversion, hence the low coverages. At the lowest temperatures N hydrogenation is shown to be rate limiting. A nice prediction of that and a discussion of the reasons why can be found in Journal of Catalysis 220, 2003, Pages 273-279.

This is an excellent point raised by the reviewer. We have modified the 2nd to last paragraph with the following sentence and included the reference:

“At higher pressure with more conversion, resulting in higher NH₃ content, the back reaction may become an important consideration, and indeed for Ru it has been theoretically predicted that the coverage of nitrogen species may become significantly higher²⁹.”

Reviewer: 2

First would we like to thank the reviewer for taking the time to read the manuscript and provide a critical review. Their comments have helped us to develop a stronger and clearer manuscript.

In the article "Operando Probing of the Surface Chemistry During the Haber-Bosch Process", by Goodwin et al., the authors use ambient pressure X-ray photoelectron spectroscopy to track the surface species and oxidation state of single crystal model catalysts for the formation of ammonia from nitrogen and hydrogen. This is a very important process in the industry, which is not fully understood, and studies like the one presented here are extremely important. The tools that the authors use are cutting-edge and appropriate for this study. However, in order to properly assess the validity of the results there are a few issues that need to be addressed. I would be happy to review this paper again after these issues have been taken care of. The paper has lots of conclusions from data that are not explicitly shown. Please include this data as xy files (or another format) in the supporting information. Others should be able to access this data to verify the statements, rather than just trusting the authors' words.

We have made a supplementary material document and included more data.

1. Text in figure 1a is too small and impossible to read.

Figure 1a has been updated for readability.

2. Figure 1b is not properly described. It looks like it is a gas flow and mass spec signal vs time plot, but that's not what it says in the caption.

We have rephrased the figure 1b caption to read:

"b, Mass spectrometer readout of mass 15 corresponding to NH₃ production as the gas ratio changes from 150 mbar pure N₂ (blue region) to 300 mbar 1:1 N₂:H₂ (green region) over Ru at 623 K. c, The enhanced mass spectrometer signals were time averaged during the interval of the 1:1 N₂:H₂ mixture to estimate the relative chemical reactivity."

3. Also in figure 1b, what are the units of the mass spec signal? Mass 15 is tracked for ammonia, but this is not explicitly said in the figure or caption. Why is there a signal before adding H₂? Signal increases from ~10 to 12 (in whatever units that is). The authors do not include the mass spec data for any other gases. It is common in mass spectrometer that there is some cross-talk between channels (especially for very small signals like this one), how do the authors know that this increase is not from cross talk with mass 2 from H₂? How does mass 28 from N₂ change when H₂ is added? Please include this data at least in the supporting information. This is critical, as some of the main conclusions of the paper are based on this data.

A more detailed description of the MS data has been added to the methods section including an explanation of background signal and overlapping signals. Furthermore, a new figure is included in the extended data to show how the background signal changes when there is no reaction occurring.

The additional text in the methods section reads as followed:

“With an instrument exposed to many gases over the years there are signals at all masses, including mass 15, before any ammonia synthesis experiment is performed, due to desorption from the chamber walls. This desorption in the first differential pumping stage is most likely coming from hydrocarbons. We therefore use the change in the mass 15 from pure N₂ to the mixture of N₂ and H₂ where ammonia is produced for determination of NH₃ production. Extended data figure 3a shows an example MS time trace where there is no NH₃ production, displaying how the background change with gas flow is within the noise of the measurement.”

4. Figure 1c. Can error bars be included in any way? This calculates a TOF based on the mass spec data. It is not clear enough how this TOF is obtained. Please include the data used to obtain the TOF in the supporting information and an example of how the data is converted into TOF.

The errors have been estimated based on the noise level shown in fig. 1b and a description of this has been included in the methods section. Since often the usage of TOF could give the impression that we have measured absolute TOFs we have changed the wording to “relative chemical reactivity” since the current set-up within the XPS measurement chamber would not allow absolute measurements.

” The errors of the relative chemical reactivity measurements have been estimated based on the noise level of the mass spectrometer signal. From the noise level, an absolute error of 0.08 in the relative chemical reactivity scale in Fig. 1c corresponds to a 95% confidence interval of the measurements. Extended data Fig. 3a shows an example of when there is no ammonia production at the lowest temperature with the least active catalyst, Fe(110) at 523 K (300 mbar, 1:1 ratio). Here we can see that extremely small ammonia production occurs and this most likely is the background level. Extended data Fig. 3b in contrast shows the same surface and experiment at the higher temperature of 673 K and clearly shows that ammonia production increases with increasing temperature.”

The data of figure 1b is directly used to determine only the relative chemical reactivity as determined in terms of the ratio between mass 15 and the sum of masses 2, 28 and 15, between different temperatures and catalytic surfaces at the same pressure and gas ratio. To clarify this the following statement was added to the methods section:

“To determine the relative chemical reactivity, the NH_3 , N_2 , and H_2 signals were monitored as the gas was switched from N_2 to $\text{N}_2:\text{H}_2$ in a 1:1 ratio (shown in Fig. 1b) to pure H_2 . The percentage of ammonia in the MS is then calculated based on the NH_3 , N_2 , and H_2 signal intensities. We then follow how this signal changes, per surface site (based on the crystal lattice), for different temperatures and single crystal surfaces at the same constant pressure and gas ratio. We thereby normalize out any variations due experiments conducted at different times through only making relative measurements. No attempt was made to estimate the absolute TOF since there are too many unknowns in the flow conditions around the sample as well as potentially stronger interactions between ammonia and the chamber walls in its path to the mass spectrometer in comparison to nitrogen and hydrogen.”

5. Given that these experiments use synchrotron light, beam effects have to be considered. The authors describe these studies in the paper, but this description is not detailed enough to understand how they did these tests, and none of the data from these tests is included. Please include the data to support your statements.

A more detailed description of the beam effects has been included in the methods section as well as a new figure in the extended data.

The additional text in the methods section reads:

” Extended data Fig. 4a shows the effect of photon flux on the nitrogen content. Note that the line of best fit shown in grey has a forced intercept to zero. Extended data Fig. 4b shows the effect of flux on nitrogen speciation, showing no change to the components of the N 1s spectra. These spectra were gathered at 500 mbar in a 1:3 gas ratio at 423 K over Ru, the conditions expected to be the most sensitive to beam damage. Equivalent tests were preformed for all catalysts.”

The new figure in the extended data is:

Extended Data Fig. 4 | Beam flux effect on surface species. **a**, The trend of N 1s intensity over Ru at 500 mbar, 423K, 1:3 N₂:H₂ gas ratio as beam flux increased with a line of best fit in grey (intercept set to zero) showing the beam does not have an appreciable effect on total coverage. **b**, The same experiment showing the N 1s spectra gathered sequentially from lowest flux to full flux showing no change in speciation with beam flux.

6. The authors state that nitrides do accumulate with time under the beam. Isn't this expected to affect the conclusions? or are these nitrides not expected to participate in the reaction? Does that mean that the beam facilitates the breaking of the NN bond?

The beam has a definitive effect to increase the rate of nitride formation under pure nitrogen. No effects of the beam were observed when any amount of hydrogen was mixed with nitrogen, when no growth of nitride occurs, see extended figure 4a. It looks like there is a time delay in the nitride formation as the N₂ is introduced, as shown in figure 2. There can be some sort of restructuring on the surface and then the diffusion occurs in the bulk. We think the diffusion and dissociation is sped up by the beam. As the only finding in the paper regarding the rate of nitride formation is that it is much slower than hydrogenation and no quantitative effort is made for the nitride growth the effects of the beam do not impact the general findings. As for the exact species of the nitrides that form and the rate that these nitrides form under pure nitrogen, these findings are affected by the beam. Extended data figure 4b shows the extent of this effect. For this reason, the manuscript does not give any weight to the rate of the nitride formation. A more detailed study of these effects and conclusions is ongoing.

A new figure is added to the extended data to show how the beam does not affect the species on the surface under reaction conditions.

The following text has been added to the methods section:

“Nitrogen spectra gathered outside of reaction conditions (i.e. without hydrogen) like those shown in fig. 2 show not only chemical changes inherent to the reaction but also accumulation of beam induced effects. The buildup of beam induced nitride formation is slower than chemical activity but is not possible to fully avoid. For this reason, no attempt is made to quantify the formation rate of the various nitrides. The main finding of the paper is the distinct lack of nitrogen on the surface during the reaction and the slow nitride formation compared to fast reduction caused by hydrogen. The beam induced nitride formation only serve to increase nitrogen formation. Therefore, the beam effects do not alter any conclusions of the activity of N_2 compared to H_2 , but rather strength the finding.”

7. For much of the data spectra are shown as a function of temperature and pressure. However, no time dependence is described. For how long was the sample exposed to these conditions? Is there any change with time on the surface?

The only data that is evolving with time is the iron data under pure nitrogen. All other spectra were gathered over the course of 3-20 hours and showed no time dependence.

The following statement was added to the methods sections to provide more clarity:

“ Spectra were gathered for between 3 and 20 hours, and no spectral changes occurred during this time when any amount of hydrogen was present. In the absence of hydrogen, time evolutions like those shown in Fig. 2 were observed.”

Reviewer: 3

First we would like to thank the reviewer for taking the time to read the manuscript and provide a critical review. Their comments have helped us to develop a stronger and clearer manuscript.

In this paper the authors use the unique POLARIS APXPS instrument to Fe(110), Fe(210), and Ru(10-13) model catalyst surfaces exposed to N₂ and N₂:H₂ mixtures at total pressures between 150 mbar – 1 bar. The data presented are quite nice and I enjoyed reading the paper. Correct labeling of figures and an occasionally more consistent discussion would have made the reading experience even better (see my detailed critics below).

The key finding of the work is that all 3 surfaces remain metallic at high temperatures of 673 K, with low surface coverage of adsorbates. Furthermore, the authors identify differences between stepped Fe and Ru surfaces and convincingly demonstrates that the rate-limiting step at 673 K is N₂ dissociation on Ru, while the hydrogenation step also is essential for the total rate on the stepped Fe surface. While the APXPS data mostly are convincing (see my detailed critics) below, I am more unsure whether the NH₃ formation are demonstrated convincingly with QMS on all studied surfaces (see also my detailed critics below).

The theme of the paper is related to recent article published articles in Science (ref. 27) and ACS catalysis (ref. 25) by the same group. In their previous work the authors studied CO₂ and CO hydrogenation over a Zn/ZnO/Cu(211) surface to mimic real CuZn-based catalyst used for methanol synthesis (ref. 27) and Fischer-Tropsch synthesis (ref. 25). Therefore, I would say that the novelty of POLARIS instrument already has been demonstrated and communicated. Thus, this cannot be used as an argument to publish in Nature.

I am not an expert in the literature of Haber-Bosch process, but as far as I can read different surface terminations have been suggested at operational conditions and also different rate determining steps have been suggested. The present article use for the first time APXPS to show that the studied surfaces at pressures close to one bar are metallic and that N₂ dissociation is the rate determining step on stepped Ru while the hydrogenation step is suggested to play a role also on Fe surfaces.

Whether the first APXPS studies performed close to 1 bar pressure or 2-3 orders of magnitude higher than conventional APXPS studies (but 2 orders of magnitude lower than real conditions) is sufficient to warrant publication in Nature is difficult for me judge. This is most likely better judged by experts in the Haber-Bosch process that have a complete and up-to-date overview of the current understanding of this process. Below follows my detailed critics which I hope the authors can use to improve their article.

If the editor decide that novelty is insufficient for publication in Nature I believe the article fits well in Nature catalysis or Nature communication.

Major points:

1. In general, I miss details for the QMS data.

It is a bit unclear from text how the results of figure 1c have been conducted. On p. 4, l. 107 – 109 the authors write. “The relative turnover frequencies at various temperatures and between single crystal catalysts were determined by calculating the percent NH₃ formed in the gas stream at a pressure of 300 mbar, shown in Fig. 1c.”. The figure caption of figure 1b mentions a gas composition of 1:1, but it is unclear from the text if all datapoints in figure 1c were conducted at 300 mbar total pressure and a 1:1 gas composition. After carefully reading the method section I understood that indeed a 1:1 gas composition was used. I encourage the authors to state this clearly in the text. The authors discuss “relative TOF in arb. units” a term I am not familiar with. As I understand the methods section the authors calculate this by first finding the relative gas composition of NH₃ and then dividing this by the number of sites on the catalyst surface. I am not sure that the general readership of high impact journals will understand the thoughts behind this. I would suggest that the authors in the text or the method section mention that the partial pressure of NH₃ is proportional to the turnover frequency, i.e. the number of reaction turnovers per second and that they then normalized this to the atomic density of Fe or Ru atoms (assuming no reconstruction of the surface).

An alternative would be to state the relative conversion per site in figure 1c where 100% refer to reaching the mass transfer limit. This is easier to understand for general readers I think. Also one avoid “arb. units”.

I assume that the authors assumed that the reaction only takes places at the front sites of their crystals? No information about this is given in the text and if so the authors need to motivate why this can be done.

Finally, I assume that the authors did not calibrate the QMS with known gas compositions and miss a sentence about this in the method section if this were the case.

Since we have complex flow conditions and unknown interactions of ammonia with the vacuum chamber walls on the path to the mass spectrometer, we didn't attempt to determine an absolute turn-over frequency. Therefore, we measured the relative reactivity by determining the mass 15 with respect to masses 28 and 2 with varying temperature and between crystal surfaces when the pressure and gas ratio is identical. Since the reviewers have maybe misunderstood that we tried to determine the absolute turn-over frequency, and so potentially also the readers, we have changed the wording to “relative reactivity”. We observe how the reactivity changes for different temperatures and between the surfaces that can be connected to the XPS data.

We added the following text in the method section.

“To determine the relative chemical reactivity, the NH₃, N₂, and H₂ signals were monitored as the gas was switched from N₂ to N₂:H₂ in a 1:1 ratio (shown in Fig. 1b) to pure H₂. The percentage of ammonia in the MS is then calculated based on the NH₃, N₂, and H₂ signal intensities. We then follow how this signal changes, per surface site (based on the crystal lattice), for different temperatures and single crystal surfaces at the same constant pressure and gas ratio. We thereby normalize out any

variations due experiments conducted at different times through only making relative measurements. No attempt was made to estimate the absolute TOF since there are too many unknowns in the flow conditions around the sample as well as potentially stronger interactions between ammonia and the chamber walls in its path to the mass spectrometer in comparison to nitrogen and hydrogen.”

2. As the authors state themselves little NH₃ is produced. Therefore, one also needs to very careful with the evidence for the NH₃ activity. On p. 4. L. 109 – 111 it is said that: “The reaction rate increases with increasing temperature and is higher for the stepped Fe(210) than the flat Fe(110) surface in agreement with previous Fe single crystal high pressure reactor studies”. Looking at figure 1c I honestly do not see much evidence for increased reaction rate on the Fe(110) surface. Maybe a little for the points at 400 dC and 450 dC, but it is very little and without any error bars, very few data points, and no information about the where the baseline is (i.e. activity at RT) it is difficult to evaluate.

We have added the MS 15 time traces for the Fe(110) at the lowest and highest temperatures demonstrating the increase in reactivity.

We have added the following extended figures 3a and 3b.

Extended Data Fig. 3 | MS time trace without ammonia production. a, The equivalent experiment as for the data shown in Fig. 1b but at 523 K over Fe(110). **b,** The equivalent experiment but at 673 K over Fe(110). Note here the time scale is far longer than in Fig. 1b to ensure sufficient statics, ensuring the relative reactivity is representative of the lack of ammonia production. Note that in **a**, there is a glitch in the mass spectrometer background as a response to the removal of N₂ but it is clear that mass 15 is decreasing in pure H₂ after the switchover.

I also miss a blind experiment on a sample holder without any crystal. Is there really zero NH₃ formation in this case at 450 dC? I would suspect this, since the sample holder most likely has an higher temperature than the single crystal surface due to the conductive heating system used. Without such a reference experiment it is difficult to be sure that the very low NH₃ activity does not come from other hot heated parts near the sample surface.

The measurement cannot be conducted without a sample with a small gap to the front cone of the instrument. However, the strong flow conditions towards the sample, creating a ring of around 2 mm of exposure on the single crystal surface, and then out towards the surrounding vacuum, make it extremely difficult for any gases to penetrate into the small gap from the chamber. Therefore no reactions occurring on the sample holder or heater can enter into the 30 μm gap to the front cone and diffuse over 1 cm against a flow of 3 l/min. Furthermore, the sample holder is cooled below approximately 350 K, far too low for any significant ammonia formation.

We have added the following text in the methods section:

“Due to the specific design of the high flow virtual cell, unwanted gas molecules originating from reactions of the sample holder or heater cannot reach the single crystal surface area that is probed by the opening into the electron spectrometer.”

3. I am confused about the pressure and temperature used for the experiments shown in figure 2. In the figure caption to figure 2 it says that spectra collection begins once a pressure of 150 mbar is reached. In line 128-129 on p. 4 the text states that the samples are exposed to 250 mbar? Finally, the extended data for figure one states 200 mbar? I have simply no idea what gas mixture that was used for the data collection in figure 2a, 2b, and 2c. Please specify this so it is clear.

We are grateful that the reviewer spotted errors in our description. The pure N₂ pressure related to figure 2 is indeed 150 mbar in accordance to the figure caption. The extended figure is related to figure 4 where the total pressure was 200 mbar and a different gas ratio of 1:3 was used.

Since the manuscript needs to be shortened and repetitions avoided, we have eliminated the repetition of the pressure in the main text since it is indicated in the figure caption.

4. Why are dC used in figure 1c, while K is used in figure 2?

We have updated the figures so all temperatures are shown in K in the manuscript.

5. I miss information in the text why temperatures of 673 K is used for the iron surfaces, while 623 K is used for the Ru surface. The authors say in line 113, p. 4 that “The maximum rate for Ru is not at the highest temperature of 723 K, as for the Fe surfaces”. I see that the highest rate for Ru is at 623 K, but line 113 seems to imply that the highest

rate on Fe surfaces is observed at 723 K. Then one should have measured at 723 K for Fe, not at 673 K? Maybe the reason is that the rate at 673 K and 723 K is almost the same on Fe? If so the authors should explain this.

The answer is less scientific but more related to the restrictions in available beamtime. These experiments were carried out during 3 beamtime periods extending over 1 ½ years. One crystal for each beamtime. We measured first Fe(110) and the relative reactivity showed little variation between 673 K and 723 K and therefore we measured the spectra at the lower temperature. For Fe(210) there is maybe a small difference between 673 K and 723 K, within the error bars. Since we had spectral data on Fe(110) at 673 K and to allow a comparison we measured Fe(210) also at the same temperature. The last beamtime was the Ru surface and then there was a drop in reactivity at high temperatures (also known from industrial catalysts) so we measured the spectra at a lower temperature. Since we have to shorten the manuscript significantly and restriction in beamtime is maybe not so interesting for the reader we think it is not a high priority to comment on this issue in the text.

6. The signal to noise ratio of figure 2a and 2b looks quite different. In fact it does not look like figure 2b is normalized to the background. Some lines look quite intense while others look weak. The authors should correct this mistake or comment why the signal to noise ratio is so different in figure 2a and 2b. Also figure 2c looks like no calibration to the background has been performed.

Not all of the data was gathered with the equivalent dwell time leading to the data on the Fe(210) appearing noisier but with finer time resolution. Again, time restricted from approved beamtime. This is therefore not due to data processing but instead available time to acquire the data. No data set has a background subtracted, only a minimum subtracted. This was chosen to allow future researchers to use the shape of the background to make deductions about the film thickness, EAL, or other concepts in development by groups like Tougaard and Shard. We think this detail is not essential enough to use limited space in the manuscript.

7. In line 153-154 the authors write: “It is interesting that a weak broad 153 feature is seen at ≈ 399 -400 eV with a binding energy position consistent with adsorbed N₂.” It took me quite some time to realize that the authors here refer to the extended figure 1. The sentence should be reformulated such that this is clear.

The text has been rewritten for clarity and now reads:

“It is interesting that a weak broad feature is seen at ≈ 399 -400 eV, with a binding energy consistent with adsorbed N₂,²⁴ see extended data Fig. 1.”

8. I like the paragraph in line 155-167 on p. 5 and I believe the authors presented solid evidence for slow N₂ dissociation and much faster H₂ dissociation and subsequent reaction at sub-bar pressures. This is good!

We are grateful for the feedback, the results stood out as significant immediately as we were gathering spectra.

9. The paragraph in line 179-191 on p. 6 convincingly shows that both Fe and Ru surfaces are metallic at elevated temperatures. This is good. However, I had initially to spend some time to understand why other pressures and gas flows were used for figure 3 than in the discussion for figure 2. I therefore encourage the authors to briefly discuss the motivation behind the experiments in figure 3 better.

The text now reads:

“Next, we address the question of oxides potentially being not reduced on Fe under operando conditions. Here the data was collected at 500 mbar, 1:3 N₂:H₂, and various temperatures. The peaks present in Fig. 3a are metallic iron at 706.5 eV, the multiplet at 707.4 eV, and the broad Fe oxide peak at 709.8 eV. The Fe(110) sample is fully reduced as the temperature reaches 523 K at 500 mbar, and the Fe(210) surface requires a higher temperature of 573 K, as seen in fig. 3b. Fe(210) needs a higher temperature due to the stronger binding of oxygen on a stepped surface. Ru is metallic at all conditions. All surfaces are in a metallic state during the HB process, as predicted by theory (Fig. 3c). Note that these measurements were gathered simultaneously with the data in Fig. 4.”

10. It looks like the panel labels in figure 4 is mixed up. In line 194-195, p. 6 it is written that: “Fig. 4abcd shows the N1s spectra of the surfaces at 673 K and different pressures with a 1:3 N₂:H₂ synthesis gas mixture”. This must be a typo? Figure 4a-d shows data recorded at 523 K. Also later in line 216 p. 7 fig. 4c should be 4g I assume. Line 249, p. 7: “Fig. 4f...” should be “Fig. 4g...” I assume. The authors should carefully check that all labels are correct in particular for figure 4.

We appreciate the careful reading by the reviewer. As part of shortening the paper we have removed this statement and the corresponding typo.

11. I do not understand the way the Ru data are discussed for figure 4. In the last paragraph on p. 6 the authors first focus on Fe spectra recorded at 673 K (discussion starts on line 195). Following this the authors briefly discuss similar data for Ru starting line 215 (even though they refer to the wrong figure. I assume fig 4c should be fig. 4g instead).

Subsequently, data recorded at 523 K are discussed. Again, Fe surfaces are discussed first (line 231, p. 7).

Finally, in line 249, p. 8 data for Ru are discussed, but only for 673 K? Where are the Ru data in figure 4c discussed? Also, there is a clear increase of the RuNH component with pressure.

The text has been clarified. The increase in the RuNH component with pressure is clear but the very low coverage has such a high error as to make a quantitative statement regarding the change in coverage uncertain.

To reflect the clear increase in signal the text has been rewritten:

“The NH signal increases with pressure, but the nitrogen coverage quantification of these results are nearly within the margin of error. If there is an increase in coverage with pressure for Ru, it may indicate that the H₂-metal interaction for Ru is weaker than for Fe, possibly leading to higher coverages at operational pressures. The adsorbed N species is much more reactive on Ru than Fe, supporting previous theoretical predictions¹⁵. “

12. I do not understand the sentence in line 312-314 on p. 9: “Since the coverage of H₂ at the reaction temperature is expected to be low, we can assume that there is no inhabitation of adsorbed hydrogen for the N₂ dissociation”. Maybe the authors mean: “Since the coverage of H₂ at the reaction temperature is expected to be low, we can assume that there is no inhabitation due to/caused by adsorbed hydrogen for/on the N₂ dissociation”

The text has been rewritten.

“Since the coverage of H₂ at the reaction temperatures is expected to be low, we can assume that there is no inhabitation of N₂ dissociation caused by the adsorbed hydrogen²⁸. ”

13. I am afraid that the discussion/conclusion for Fe surfaces starting line 274, p. 8 and ending line 326, p. 9 is difficult to read for readers of high impact journals. In fact, I am not sure that I understand the take-home message myself. For example, line. 311-312 on p. 9 say that “...directly implies that the N₂ dissociation step is slower than the hydrogenation step of the adsorbed N (here the authors discuss the Fe)” while it a few lines above is written that: “we observe adsorbed N that are more populated on the more strongly bonded stepped crystal, and the hydrogenation step 3 also partly controls the rate”. To me it is unclear what controls the rate on the Fe surfaces and the text reads like different things are said at different places. Another example is the discussion of figure 2 where it also is highlighted that H₂ is very quickly removes N-species from the surface.

The text has been rewritten for clarity and reads:

“ On Fe it is well established that the rate limited steps is the molecular dissociation⁷⁻⁹, supported by the correlation between the NH₃ production rate and the N₂ dissociative sticking coefficient for the different single crystal surface facets.^{9,28} However, the results here show that, at all temperatures, a factor of around 100 times

higher population of adsorbates is observed in comparison to the stepped Ru surface, and we no longer can postulate that the reaction proceeds with a high rate after the molecular dissociative steps. Furthermore, there are no signs of molecularly adsorbed N₂ even at the lowest temperatures, indicative of a much higher rate of step 1b. Above 573 K, we observe adsorbed N that is more populated on the stepped crystal, indicating that the hydrogenation step 3 also partly controls the rate.¹²

The coverage of N species on the Fe surfaces lowers with increasing total pressure at a constant N₂:H₂ ratio, implying that the N₂ dissociation step is slower than the hydrogenation step¹⁰. Most likely the coverage of adsorbed H increases with pressure resulting in faster hydrogenation. Since the coverage of H₂ at the reaction temperatures is expected to be low, we can assume that there is no inhabitation of N₂ dissociation caused by the adsorbed hydrogen²⁸.

14. In line 331 p. 10 the authors write: “With improved instrument sensitivity allowing high quality spectra to be measured in seconds or minutes (not as here many hours) will provide...” sounds like claim without any support. As I understand it the instrument sensitivity of POLARIS is limited by the incoming flux and the electron scattering in the gas phase. For such a statement the authors need to discuss that it is feasible to reach 60 or 3600 higher overall instrument sensitivity.

The reviewer is correct that such claims are not suited for this paper. The statement has been removed. In brief for the reviewer, other than flux there are several means to improve signal quality, including geometric optimization, clever pressure jumping experiments, and new aperture designs in the front cone of the electron spectrometer.

Minor parts:

15. “N1s” and all other core levels should be written with a space between the element and the core level (“N 1s”)

Corrected.

16. Some of the figure text is way to small to be read on a standard printout of the article. This is for example the case for embedded text in figure 1a, many captions, etc.

The figures have been updated.

Reviewer Reports on the First Revision:

----- Referees' comments -----

Referee #2 (Remarks to the Author):

This paper describes the reaction of $N_2 + H_2$ to form NH_3 using AP-XPS and MS at pressures higher than ever achieved before with AP-XPS. This technical achievement is remarkable. However, I am afraid the data in its current form do not adequately support the conclusions obtained.

Some of my prior comments have been only partially addressed.

1. Top right of Figure 1a still shows an XPS spectrum that is impossible to read.
2. This is related to my prior comment #2, the caption in Figure 1b still makes no reference to the flow in the y-axis; I assume this flow is related to the partial pressure described in the caption?
3. Related to my prior comment #3:

Extended Figure 3 was added to support the fact that the mass spec signal (still no units) is truly from ammonia production. I don't see how this figure supports this statement. In figure 1b, the increase in this signal is about 20%, and the fluctuations in extended figure 3 are also in that range. Extended figure 3 is also not properly described; what is the shaded orange region?

One of the things I asked in the comment is to include the MS traces from other species involved in the process, such as N_2 and H_2 , to check if the change in the signal is not from cross-talk between channels in the mass spec when introducing H_2 . This request was ignored entirely.

Additionally, m/z 16 and 17 should be included since 17 gives by far the most intense signal. I saw the authors claim these overlap with signals from water, but the authors are looking at changes relative to the background signal (which is already large at 15 anyway), so they would be looking at increases with respect the water background level. Either way, it is suspicious that none of this data has been included in the supporting information, even after being requested by a reviewer.

Again, some of the main points of this paper are based on this unreliable data...

4. Related to my prior comment #4 and the authors' response:

The authors state that an absolute error of 0.08 was obtained from the noise level. However, there's no description of how this error is obtained, and it is not clear what the units are. Please describe this further.

What does a relative activity scale mean? Please describe in detail how this is calculated. The exact formula used should be reported. I understand, based on the text, that it is calculated from mass spec signals from NH_3 , N_2 , and H_2 . Still, a formula is obviously used for the calculation, normalization, etc, so this should be reported along with the currently missing mass spec data.

5. Related to my prior comment #5 on beam effects:

Are there mass spectra for the various involved species to show if there are any changes with the beam on vs off?

6. Related to my prior comment #6 on beam-induced N_2 dissociation and the authors' response:

While almost no N is shown on the surface by XPS under N_2+H_2 , the fact that the authors claim NH_3 is produced must mean that N_2 and H_2 are dissociating on the surface. The fact that the reaction is fast enough that N species are not visible doesn't mean that there's no beam effect on the reaction. The fact that the beam induces N_2 dissociation likely does affect the overall reaction. Again, the mass spec data with the beam on and off should shed light on this.

In fact, one of the main conclusions of this paper is that N_2 dissociation is the rate-limiting step in

the case of Ru!

7. Related to the response to my prior comment #7: quoting the authors' response "All other spectra were gathered over the course of 3-20 hours and showed no time dependence."

Just to clarify, are the authors saying that they run the experiment under the N₂ + H₂ mixture for up to 20 hours, and no changes in the XPS nor the mass spec data were observed during this time period? Can the authors show this data? (XPS and MS vs time for this 20 hours period that the authors state show no time dependence).

Referee #3 (Remarks to the Author):

I read through the reply to my questions and am pleased to find that the authors carefully replied all my comments. I have no more feedback that potentially can improve the article.

Author Rebuttals to First Revision:

Reviewer: 2

First, we would like to apologize to the reviewer that we were not careful enough in reading the comments in the previous reports and therefore missed some of the questions. Secondly, we appreciate that the reviewer has raised again these points since addressing them makes the paper clearer for all readers.

This paper describes the reaction of $N_2 + H_2$ to form NH_3 using AP-XPS and MS at pressures higher than ever achieved before with AP-XPS. This technical achievement is remarkable. However, I am afraid the data in its current form do not adequately support the conclusions obtained.

Some of my prior comments have been only partially addressed.

1. Top right of Figure 1a still shows an XPS spectrum that is impossible to read.

We have increased the font size of the XPS spectrum in Fig. 1a.

2. This is related to my prior comment #2, the caption in Figure 1b still makes no reference to the flow in the y-axis; I assume this flow is related to the partial pressure described in the caption?

We have updated the figure caption to Fig. 1b.

It now reads:

Mass spectrometer readout of masses 15 and 16 corresponding to NH₃ production as the gas ratio changes from 150 mbar pure N₂ (blue region showing flow) to 300 mbar 1:1 N₂:H₂ (green region showing flow) over Ru at 623 K. Note that the flows of the gases are shown as the filled blocks plotted on the left axis.

3. Related to my prior comment #3:

Extended Figure 3 was added to support the fact that the mass spec signal (still no units) is truly from ammonia production. I don't see how this figure supports this statement. In figure 1b, the increase in this signal is about 20%, and the fluctuations in extended figure 3 are also in that range. Extended figure 3 is also not properly described; what is the shaded orange region?

We have extensively changed Extended data Fig. 3, and renumbered it to be Extended data Fig. 4, to respond more directly to the reviewer's concerns. We also have included in Extended data Fig. 4 (formerly 3) the beam's effect on the ammonia signal. Please see our response to points 5 and 6.

In order for the POLARIS instrument to reach pressures close to atmospheric, it is necessary to have a high gas flow. Therefore, the ammonia signal above the background will be weak compared to a stationary set-up where the rate can be measured more accurately. Here we sought to determine that we are semi-qualitatively producing ammonia in direct connection to the XPS measurements. We have written in the Methods section that the signal is detected based on time integration before and after the syngas is introduced in order to reduce the effects of the

noise. We are confident that we are detecting ammonia, supported by the temperature dependence of the results following known trends. Every other variable is constant, and the sample is far away from the mass spectrometer, so the sample temperature would not affect the mass spectrometer signal.

We have further clarified this in the Methods section and the text reads:

“As the MS is highly sensitive, there is signal before any experiment from the chamber at all masses including masses 15 and 16, most likely due to hydrocarbons. With the high flow conditions required to establish the pressure for the XPS measurements, the amount of ammonia in the gas stream into the mass spectrometer is small and the signal becomes noisy. Therefore, in order to make a more accurate measurement of the NH_3 production, time integration was done between the background level in pure N_2 and that of pure H_2 . The background was subtracted from the time integration during ammonia production. Extended data Fig. 4a shows an example MS time trace where there is negligible NH_3 production, showing how the background change with gas flow is within the noise of the measurement and therefore requires the time integration. Due to the specific design of the high flow virtual cell, unwanted gas molecules originating from reactions of the sample holder or heater cannot reach the single crystal surface area that is probed by the opening into the electron spectrometer. Thereby, all measurement conditions are constant. The increase in ammonia production at higher temperatures is as expected according to refs.^{9, 21, 22}, providing further confidence that ammonia is produced.”

We have updated Extended data Fig. 4 (formerly 3) to clarify that the yellow region is when N_2 is removed and only pure H_2 is being flowed. Below is the new figure:

One of the things I asked in the comment is to include the MS traces from other species involved in the process, such as N₂ and H₂, to check if the change in the signal is not from cross-talk between channels in the mass spec when introducing H₂. This request was ignored entirely.

Additionally, m/z 16 and 17 should be included since 17 gives by far the most intense signal. I saw the authors claim these overlap with signals from water, but the authors are looking at changes relative to the background signal (which is already large at 15 anyway), so they would be looking at increases with respect the water background level. Either way, it is suspicious that none of this data has been included in the supporting information, even after being requested by a reviewer.

Again, some of the main points of this paper are based on this unreliable data...

The updated Extended data Fig. 4 (formerly 3) shows masses 2, 28, 14, 15, 16, and 17, as requested by the reviewer. Furthermore, the new Extended data Fig. 3, also shows the MS traces for masses 15, 16, 17, and 18 as part of our response to this point as well as point 7.

Mass 16 has now also been included in the signal for ammonia to reduce concerns regarding cross talk between mass fragments. While mass 15 could be slightly affected by cross talk wherein N and H combine within the MS, the possibility of N and two H atoms combining is much too low to account for the signal present. Furthermore, we believe that with the changes to Extended data Fig. 3 we can show that since multiple mass fragments respond to temperature, the signal must originate at the sample. The sample is the only place that is heated and no other variable is changed beyond temperature and, therefore, activity.

To explain why mass 17 is excluded, a new figure: Extended data Fig. 3 has been included in the Methods section. The figure shows how masses 15, 16, and 17 evolve over a long aquation time. From this, we can see that mass 15 has the least contamination from water; 16 can be included with appropriate considerations, but mass 17 has major complications. These complications are shown by the fact that after processing mass 17 falls below zero using standard values for the cracking pattern of water, but over the course of the measurement increases. To include mass 17 in the signal of ammonia would require an ad-hoc ionization pattern of water, and since these experiments were performed over the course of 3 years the ionization pattern would have to be determined not by use of a calibration gas but instead by the background signals prior to experiments. The process needed to include mass 17 is fully within the scope of the research but would introduce a very significant bias into already noisy data. For this reason, it was chosen to simply exclude mass 17 and use the data that would require the least processing to show catalytic activity.

The following statements have been added to the Methods section:

Within the MS, it is possible for cross talk between channels or other instrumental errors to affect the signal. This is particularly true when the signal is very near the noise level, as in the work presented herein. Extended data Fig. 5 shows the masses 15, 16, 17, and 18. Mass 17, corresponding to ammonia, is strongly affected by water production from H₂ interaction with the chamber walls and the mass spectrometer itself, making quantitative analysis impossible. To decrease the possibility of the ammonia signal originating from instrumental errors, both

masses 16 and 15 are included in the signal of ammonia. As discussed above, mass 17 is not included due to the large water signal. Extended data Figures 3a and 3b show the effect of both processing and long acquisition times. The sample is Fe(210), 423K, 500 mbar, 1:3 N₂:H₂ ratio. Here we can see that both 15 and 16 AMU are constantly above the background signal from hydrocarbons or water. Meanwhile, mass 17 is not, due to the strong overlap of OH and NH₃ masses. Note that the background signal removed at this point in the processing does not account for all of the background signals. As described above, to determine the relative chemical activity, the signal of ammonia (masses 15 and 16) above the background signal in either pure N₂ or pure H₂ is taken. That ammonia signal is then compared to the total signal within the MS over the same time period. By this method, the plotted data does not remove all of the background signals, yet when the data is processed for relative chemical activity, the entirety of the background is removed.

Extended Data Fig. 3 | Stability of long-time acquisitions. **a**, The set of MS data prior to any processing gathered over a two hour window, the trend in masses 16,17, and 18 are due to the slow improvement of the vacuum conditions. **b**, The same data as in 5a with processing as described in the Methods section. **c**, XPS spectra gathered simultaneously normalized to background signal, and **d**, the sum of the spectra with the fitted peaks displayed according to the color scheme used throughout the paper.

4. Related to my prior comment #4 and the authors' response:

The authors state that an absolute error of 0.08 was obtained from the noise level. However, there's no description of how this error is obtained, and it is not clear what the units are. Please describe this further.

What does a relative activity scale mean? Please describe in detail how this is calculated. The exact formula used should be reported. I understand, based on the text, that it is calculated from mass spec signals from NH₃, N₂, and H₂. Still, a formula is obviously used for the calculation, normalization, etc, so this should be reported along with the currently missing mass spec data.

A more complete description of the error is now included in the Methods section:

The error of the relative chemical activity is estimated based on the signal to noise ratio of the background of the ammonia signal. Part of the calculation is to subtract the background, seen in Figure 1b, between time 15-23 min.; the fluctuations in the background can have a significant effect on the calculation of ammonia content. To ascertain the estimated error, the 95% confidence interval of the noise average and standard deviation over the collected time were introduced as an error source in the equation for relative chemical activity. Since the background signal and noise are similar for all experiments, the estimated error introduced is also similar. The relative chemical activity is meant to be a semi-quantitative description of the abundance of ammonia, only as a comparative description of these similar systems, and to demonstrate that the trends follow previous more absolute activity measurements.

We have also added error bars to Fig. 1c-

The following equation and description have been added to the Methods section to describe the relative chemical activity.

The relative chemical activity (RCA) was calculated using the following equation. Time averaged NH₃% was calculated from the amount of signal from ammonia as described above per total signal from the MS. Volume (V) of gas is the total volume of gas used during the measurement, pressure (P) over the sample, temperature (T) of the sample, the gas constant (R), time is the duration of the time when ammonia could have been produced, A_n is Avogadro's number, and sites is the number of active sites under the high pressure area. Finally, the highest activity on any surface is a normalization to the maximum of any surface. The normalization is

to account for systematic errors, such as the fact that most of the volume of gas used does not pass over the sample, or the fact that not all sites in the high pressure region under the front cone would be probed by the MS.

$$RCA = \frac{\overline{NH_3\%} * V * P}{sites * Time * T * R * A_n} / \text{Highest activity on any surface}$$

5. Related to my prior comment #5 on beam effects:

Are there mass spectra for the various involved species to show if there are any changes with the beam on vs off?

Extended data Fig. 4 (formerly 3) now includes the effect of the beam on ammonia production. Extended data Fig. 4c (formerly 3c) shows all of the mass fragments when the beam is on and off. The following has been added to the Methods section:

To determine if the beam has any effect on the MS findings, two experiments of gas switching (where the sample was exposed to pure N₂, then 1:1 N₂:H₂, then pure H₂) were performed with x-ray light, and without. Extended data Figs. 4 b and c show the mass fragments of all relevant species for these experiments. For this data, the sample was Fe(110) at 673 K. Extended data Fig. 4 b shows the measurement with beam, while c was collected without beam. While there are some minor differences in the ammonia signal between the two datasets, none of the changes are what would be expected from beam effects. It is expected that beam effects within the MS would strictly cause an increase or decrease in ammonia signal. The change in the relative chemical activity between the test done with and without beam is approximately 2% and well within the error of the experiment. From this, it is clear that the x-ray beam does not have any effect on the MS findings.

Extended Data Fig. 4 | MS time trace with and without ammonia production. a, The equivalent experiment as for the data shown in Fig. 1b, but at 523 K over Fe(110) with the complete set of mass fragments shown prior to any data processing, as well as the ammonia signal with all of the data processing. **b,** The equivalent experiment but at 673 K over Fe(110). Note here the time scale is far longer than in Fig. 1b to ensure sufficient statistics, ensuring the relative reactivity is representative of the lack of ammonia production. Note that in **a**, there is a glitch in the mass spectrometer background as a response to the removal of N₂ but it is clear that mass 15 is decreasing in pure H₂ after the switchover. **c,** The equivalent experiment as **b**, but performed without any x-rays on the sample.

6. Related to my prior comment #6 on beam-induced N₂ dissociation and the authors' response:

While almost no N is shown on the surface by XPS under N₂+H₂, the fact that the authors claim NH₃ is produced must mean that N₂ and H₂ are dissociating on the surface. The fact that the reaction is fast enough that N species are not visible doesn't mean that there's no beam effect on the reaction. The fact that the beam induces N₂ dissociation likely does affect the overall reaction. Again, the mass spec data with the beam on and off should shed light on this.

In fact, one of the main conclusions of this paper is that N₂ dissociation is the rate-limiting step in the case of Ru!

The response from the previous point addresses the reviewer's questions. The level of ammonia production does not show any changes with or without the beam as seen in the new Extended data Figs. 4b and 4c (formerly 3b). Furthermore, Extended data Fig. 5 (formerly 4) shows that the atomic N XPS signal increases linearly with the beam flux giving evidence that the used beam intensity doesn't influence the measurements. If that were the case, there should be a deviation from the linear dependence, and the N1s intensity should grow more rapidly at a higher flux. Another piece of evidence is given in the nitride description in the main text: "At temperatures below 523 K, no nitride formation is observed". If the beam were inducing the N₂ dissociation when H₂ is also present on the surface we would then see nitride growth at lower temperatures when the thermal reaction is no longer active.

7. Related to the response to my prior comment #7: quoting the authors' response "All other spectra were gathered over the course of 3-20 hours and showed no time dependence."

Just to clarify, are the authors saying that they run the experiment under the N₂ + H₂ mixture for up to 20 hours, and no changes in the XPS nor the mass spec data were observed during this time period? Can the authors show this data? (XPS and MS vs time for this 20 hours period that the authors state show no time dependence).

We are sorry for the confusion in our response to the last round of reviews. We understood the question to be in regard to the time requirements to perform the experiment, including preparation, equilibration, alignment, and so on. No spectrum was gathered for more than 5

hours, with typical aquations taking 30 mins to 2 hours. The new Extended data Fig. 3 shows the XPS signal intensity and MS traces (for all mass fragments 15, 16, 17, and 18) over two hours, specifically Fe(210), 423 K, and 500 mbar.

The Methods section has been modified to read:

Individual spectra were gathered for 30 to 300 minutes with no decreable spectral changes when hydrogen was present in the gas phase. Extended data Fig. 3 shows an example time interval of 2 hours over Fe(210) at 423K and 500 mbar in 1:3 N₂:H₂ gas mixture. Extended data Figs. 3a and 3b show the data over this time for mass fragments 15, 16, 17, and 18, with and without processing. Extended data Fig. 3c shows the XPS spectra evolution with time and Extended data Fig. 3d shows the time averaged results. From these, it is clear there the only changes observed with time are the decrease in water signal due to the slow improvement of vacuum conditions under constant hydrogen conditions.

Reviewer Reports on the Second Revision:

Referees' comments:

Referee #2 (Remarks to the Author):

The authors have addressed my prior comments.